# MGPATH: A Vision-Language Model with Multi-Granular Prompt Learning for Few-Shot Whole Slide Pathology Classification

**Anh-Tien Nguyen**                                                           *anh.t.nguyen@uni-giessen.de*
*Institute for Predictive Deep Learning for Medicine and Healthcare, Justus Liebig University Giessen, Germany*
*Department of Medical Informatics, University Medical Center Gottingen, Germany*

**Duy Minh Ho Nguyen**[*]
*Max Planck Research School for Intelligent Systems (IMPRS-IS), Germany*
*University of Stuttgart, Germany*
*German Research Center for Artificial Intelligence (DFKI), Germany*

**Nghiem Tuong Diep**[*] **& Trung Quoc Nguyen & Daniel Sonntag**
*German Research Center for Artificial Intelligence (DFKI), Germany*

**Nhat Ho**
*The University of Texas at Austin, USA*

**Jacqueline Michelle Metsch & Miriam Cindy Maurer & Hanibal Bohnenberger**
*University Medical Center Gottingen, Germany*

**Anne-Christin Hauschild** [†]                              *anne-christin.hauschild@uni-giessen.de*
*Institute for Predictive Deep Learning for Medicine and Healthcare, Justus Liebig University Giessen, Germany*

**Reviewed on OpenReview:** *https://openreview.net/forum?id=u7U81JLGjH*

## Abstract

Whole slide pathology image classification presents challenges due to gigapixel image sizes and limited annotation labels, hindering model generalization. This paper introduces a prompt learning method to adapt large vision-language models for few-shot pathology classification. We first extend the Prov-GigaPath vision foundation model, pre-trained on 1.3 billion pathology image tiles, into a vision-language model by adding adaptors and aligning it with medical text encoders via contrastive learning on 923K image-text pairs. The model is then used to extract visual features and text embeddings from few-shot annotations and fine-tunes with learnable prompt embeddings. Unlike prior methods that combine prompts with frozen features using prefix embeddings or self-attention, we propose multi-granular attention that compares interactions between learnable prompts with individual image patches and groups of them. This approach improves the model's ability to capture both fine-grained details and broader context, enhancing its recognition of complex patterns across sub-regions. To further improve accuracy, we leverage (unbalanced) optimal transport-based visual-text distance to secure model robustness by mitigating perturbations that might occur during the data augmentation process. Empirical experiments on lung, kidney, and breast pathology modalities validate the effectiveness of our approach; thereby, we surpass several of the latest competitors and consistently improve performance across diverse architectures, including CLIP, PLIP, and Prov-GigaPath integrated PLIP. We release our implementations and pre-trained models at this https://github.com/HauschildLab/MGPATH.

[*] Equal second contribution. [†] Corresponding author

# 1 Introduction

Whole slide imaging (WSI) (Niazi et al., 2019) has become essential in modern pathology for capturing high-resolution digital representations of entire tissue samples, enabling easier digital storage, sharing, and remote analysis (Pantanowitz et al., 2011). Unlike conventional methods that depend on examining slides under a microscope, WSI provides faster, detailed structural and cellular insights essential for disease diagnosis across multiple tissue layers, which is particularly valuable in cancer screening (Barker et al., 2016; Cheng et al., 2021). Nevertheless, WSIs are massive images, often containing billions of pixels (Farahani et al., 2015; Song et al., 2023), making detailed annotations and analysis difficult and expensive. To tackle these challenges, machine learning techniques incorporating few-shot and weakly supervised learning have been developed (Madabhushi & Lee, 2016; Li et al., 2023; Lin et al., 2023; Ryu et al., 2023; Shi et al., 2024). Among these, *multiple instance learning* (MIL) and *vision-language models* (VLMs) have gained particular attention for their ability to effectively manage limited annotations and interpret complex whole-slide pathology images.

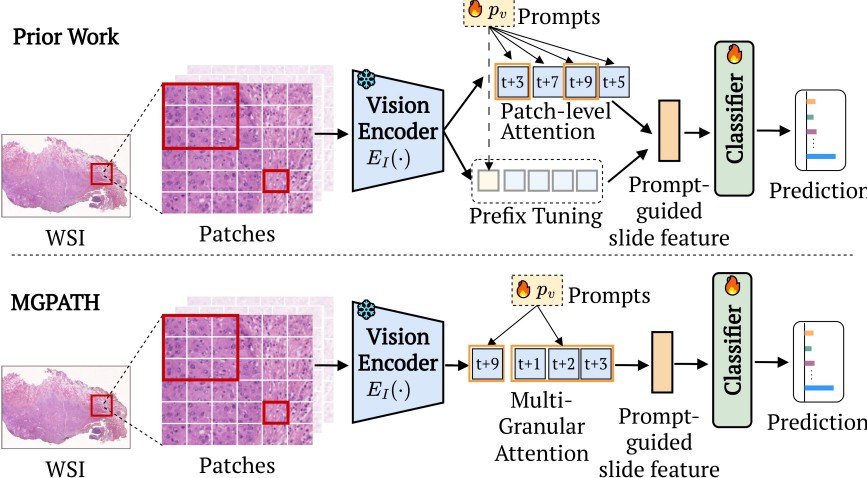

Figure 1: Unlike previous methods that add prompts at prefix positions or patch-level attention - disrupting structural correlations - our MGPATH framework integrates prompts at both regional and individual patch levels (multi-granular attention). The tokens $t+1$, $t+2$, and $t+3$ represent the spatially adjacent patches within a larger tissue region (highlighted by the larger orange box), which together contribute to the model's diagnostic decision. In contrast, $t+9$ represents a more distant or isolated patch from a different region.

In *MIL* (Ilse et al., 2018; Xu et al., 2019a; Li et al., 2023; Lin et al., 2023; Tang et al., 2023; Shi et al., 2023), each WSI is first divided into smaller patches or instances. These instances are extracted feature embeddings using pre-trained vision encoders before being grouped into a "bag", i.e., a whole slide-level representation for the entire WSI. The MIL model mainly focuses on learning ensemble functions to identify patterns in specific patches, contributing to the overall label prediction for each bag (e.g., cancerous or non-cancerous), hence reducing the need for detailed annotations. Nonetheless, these methods often struggle to select relevant patches due to complex correlations and tissue variability (Gadermayr & Tschuchnig, 2024; Qu et al., 2024b). To overcome those obstacles, VLMs (Lu et al., 2023; Huang et al., 2023; Ikezogwo et al., 2024; Shi et al., 2024) have emerged as a promising solution, combining slide-level visual features with textual descriptions to enrich contextual understanding and support predictions in sparse data scenarios with approaches such as zero-shot learning (Xu et al., 2024; Ahmed et al., 2024). Specifically, VLMs incorporate multi-scale images (Shi et al., 2024; Han et al., 2024), permitting the extraction of global and local WSI features at different resolutions. To adapt the pre-trained vision-language model efficiently, prompt learning (Zhou et al., 2022b; Gao et al., 2024) is employed where learnable prompts are treated as part of the input text to guide the model, and contextual prompts (Li & Liang, 2021; Yao et al., 2024) are integrated into feature embeddings using a self-attention mechanism (Vaswani, 2017). Despite their strong classification performance across diverse tasks, these approaches still encounter certain limitations.

*First*, (i) adapting prompt learning with frozen visual features often neglects the hierarchical relationships among learnable prompts and the visual features they interact with - specifically, *the multi-granular attention between prompts to individual patches and groups of patches.* This limitation lessens the model's ability to capture interdependence across distinct scales — from fine-grained local features to broader contextual information, leading to less accurate comprehension of complex patterns in pathology images. *Second*, (ii) many VLMs rely on the `CLIP` architecture (Radford et al., 2021), which was not explicitly pre-trained on pathology images, thereby limiting its adaptability in few-shot settings, especially when the architecture is primarily frozen and prompt learning is applied. While there exist recent works that have incorporated `PLIP` (Huang et al., 2023), a model pre-trained on 200k pathology image-text pairs curated from Twitter and showed significant improvements, an open question remains whether scaling pre-training to millions or billions of pathology-specific samples could further boost performance. *Lastly*, (iii) most VLM models for whole-slide pathology rely on cosine similarity to align visual and textual features. This metric, however, can struggle with multiple text descriptions for sub-regions (Chen et al., 2023) and with augmented data perturbations (Nguyen et al., 2024b), as it lacks the precision to capture fine-grained alignments between varied image-text pairs.

In this work, we present `MGPATH`, a novel vision-language model (VLM) designed to address the challenges of whole-slide pathology classification. Unlike existing approaches, our method integrates large-scale pathology pre-training with multi-granular prompt learning to capture both fine- and coarse-grained tissue characteristics. To enhance alignment between visual and textual representations, we adopt a parameter-efficient strategy that scales to millions of image–text pairs without requiring full model retraining. Next, we incorporate optimal transport (OT) method to measure the distance between prompt-fused visual embedding and multiple text prompts, providing flexibility in aligning heterogeneous data distributions.

Extensive evaluations on three benchmark datasets and across diverse backbones demonstrate that `MGPATH` consistently outperforms both multiple instance learning (MIL) and recent VLM approaches, achieving significant gains in F1, AUC, and accuracy over state-of-the-art baselines. Our contributions are summarized as follows:

- We propose `MGPATH`, a parameter-efficient vision–language model for whole-slide pathology that leverages large-scale pathology pretraining with lightweight adaptor-based alignment.

- We introduce a multi-granular prompt learning framework with hierarchical attention, enabling effective integration of fine- and coarse-grained contextual information from whole-slide images.

- We incorporate optimal transport to align prompt-fused visual embeddings and multiple textual prompts, providing robustness to align heterogeneous data distributions in few-shot scenarios.

- We evaluate `MGPATH` across three pathology benchmarks and multiple backbones, consistently outperforming state-of-the-art MIL and VLM baselines, with notable improvements in F1, AUC, and accuracy.

## 2 Related Work

### 2.1 Large-scale Pre-trained Models for Pathology

Recent advancements in large-scale pre-trained models for pathology can be broadly classified into two categories. *Vision models*, such as `Virchow` (Ikezogwo et al., 2024), `Hibou` (Nechaev et al., 2024), `UNI` (Chen et al., 2024), and `Prov-GigaPath` (Xu et al., 2024) leverage massive pathology image datasets to learn robust visual representations. Among these, `Prov-GigaPath` stands out as the largest model, trained on 1.3 billion pathology image patches, and excels in resolving complex tissue patterns at high resolution. On the other hand, *vision-language models* (VLMs) like `PLIP` (Huang et al., 2023) (trained 200K image-text pairs), `CONCH` (Lu et al., 2024) (1.17M), or `QUILTNET` (Ikezogwo et al., 2024) (1M), integrate visual and textual information to enhance contextual understanding and improve pathology slide interpretation. In contrast, our `MGPATH` combines the strengths of both approaches by using a *parameter-efficient adaptor* to link `Prov-GigaPath` (the largest pre-trained vision encoder) with a text encoder from VLMs like `PLIP` or `CONCH`, leveraging both rich visual features and semantic textual embeddings. Although we use the `PLIP` text encoder in our experiments due to its common use in baselines, the method can be extended to other larger pre-trained text models.

## 2.2 Few-shot learning in WSI

MIL treats a WSI as a bag of patches and aggregates these instances into a bag of features, with early methods using non-parametric techniques like mean or max pooling. However, since disease-related patches are rare, these methods can overwhelm useful information with irrelevant data. To address this, attention-based methods, graph neural Networks (GNNs), and Transformer-based methods have been introduced (Lu et al., 2021; Chen et al., 2021; Ilse et al., 2018; Li et al., 2021; Shao et al., 2021; Zheng et al., 2022). In contrast, VLMs have gained popularity through contrastive learning, aligning image-text pairs to enhance performance on a variety of tasks. While collecting large-scale pathology image-text pairs remains challenging, models like MI-Zero, `PLIP`, and `CONCH` have been trained on hundreds of thousands to over a million pathology image-text pairs (Lu et al., 2023; Huang et al., 2023; Lu et al., 2024). Some approaches also integrate multi-magnification images and multi-scale text to mimic pathologists' diagnostic processes, especially for detecting subtle abnormalities (Shi et al., 2024; Han et al., 2024). Our `MGPATH` extends on the VLMs strategy by further *amplifying the benefits of using a large pre-trained pathology* VLM model and introducing a new *parameter-efficient multi-granular prompt learning* to adapt these models to few-shot settings.

## 2.3 Prompt Learning for Vision-Language Adaptations

Prompt tuning is proposed to transfer large pre-trained model task-specific downstream tasks and has shown strong results in multimodal models like CLIP. Rather than design a heuristic template, several methods like `CoOp` (Zhou et al., 2022b), `CoCoOp` (Zhou et al., 2022a), or `MaPLe` (Khattak et al., 2023) among others (Rao et al., 2022; Shu et al., 2022) have allowed models to determine optimal prompts from multiple perspectives, such as domain generalization (Ge et al., 2023; Yao et al., 2024), knowledge prototype (Zhang et al., 2022b; Li et al., 2024), or diversity (Lu et al., 2022; Shu et al., 2022). However, these approaches focus on natural images and do not address the unique challenges of whole-slide pathology images, which require multi-scale and structural contextual information. While a few current methods typically integrate prompts with frozen visual features via self-attention (Shi et al., 2024; Qu et al., 2024a), these approaches might struggle with the complex relationships in WSIs. Our solution introduces multi-granular prompt learning, bridging *attention* on both *individual image patches* and *spatial groups* to better align with the hierarchical structure of WSI data.

# 3 Methods

This section presents `MGPATH`, a parameter-efficient vision–language framework for whole-slide pathology classification. We bridge the `Prov-GigaPath` visual encoder (Xu et al., 2024) - pretrained on 1.3B pathology patches—and the `PLIP` text encoder (Huang et al., 2023) - trained on ∼200K image–text pairs—using lightweight adaptor modules optimized with a contrastive objective. To strengthen vision–text alignment without updating either backbone, we collect an additional 923K pathology image–text pairs from ARCH (Gamper & Rajpoot, 2021), PatchGastricADC22 (Tsuneki & Kanavati, 2022), and Quilt-1M (Ikezogwo et al., 2024) and train only the adaptors via cross-alignment (Gao et al., 2024; Cao et al., 2024). This yields a large-scale aligned representation while remaining highly parameter-efficient. To our knowledge, `MGPATH` is the first parameter-efficient VLM for pathology trained at this data scale (923K pairs) on top of a 1.3B-patch visual pretraining.

Next, we leverage these pre-trained models for few-shot WSI tasks by introducing *multi-granular prompt learning*. First, descriptive text prompts are generated for image patches at dual magnification (low/high) using a frozen large language model (LLM) and augment them with multiple learnable prompt tokens per magnification, capturing diverse subregional patterns (Han et al., 2024; Shi et al., 2024; Qu et al., 2024a). Then, we integrate the learnable prompts with frozen visual features at two granularities: (i) patch-level attention over all patches and (ii) region-level attention over spatial groups built from patch coordinates via message passing. As illustrated in Figure 1, unlike previous methods, which typically score tokens independently, based on their individual relevance scores to the prompts ($t + 3$ and $t + 9$). Our method explicitly incorporates both regional attention adjacent, semantically linked tokens ($t + 1$, $t + 2$, $t + 3$), while preserving patch-level attention for isolated yet discriminative tokens ($t + 9$). This results in richer prompt-guided slide features that better capture the underlying pathology. Concretely, we represent image patches from each WSI as a spatial graph, using bounding-box coordinates to enable region-level aggregation through message passing along local connections. This spatial structure is encoded as tokens within the *Key-Value* matrices, which interact with *Query* matrices derived from prompt embeddings. By directing

attention from Query to Key-Value matrices across both patch and region levels, our approach effectively captures hierarchical information, enriching feature representation and selectively emphasizing features across diverse tissue areas. Finally, instead of cosine similarity, we align prompt-fused visual summaries to class prompts using an optimal transport (OT) distance (Nguyen et al., 2021; Pham et al., 2020; Séjourné et al., 2023; Chen et al., 2023; Dong et al., 2023; Nguyen et al., 2024b; Zhan et al., 2021), which is robust to noisy augmentations and partial text–region overlap - both common in WSIs.

Figure 2 provides an overview of the key steps in our method. We next detail the adaptor-based vision–text alignment (Sec. 3.1), the construction of multi-magnification descriptive prompts (Sec. 3.2), granularity-aware visual prompting (Sec. 3.3), and the OT-based alignment objective (Sec. 3.4).

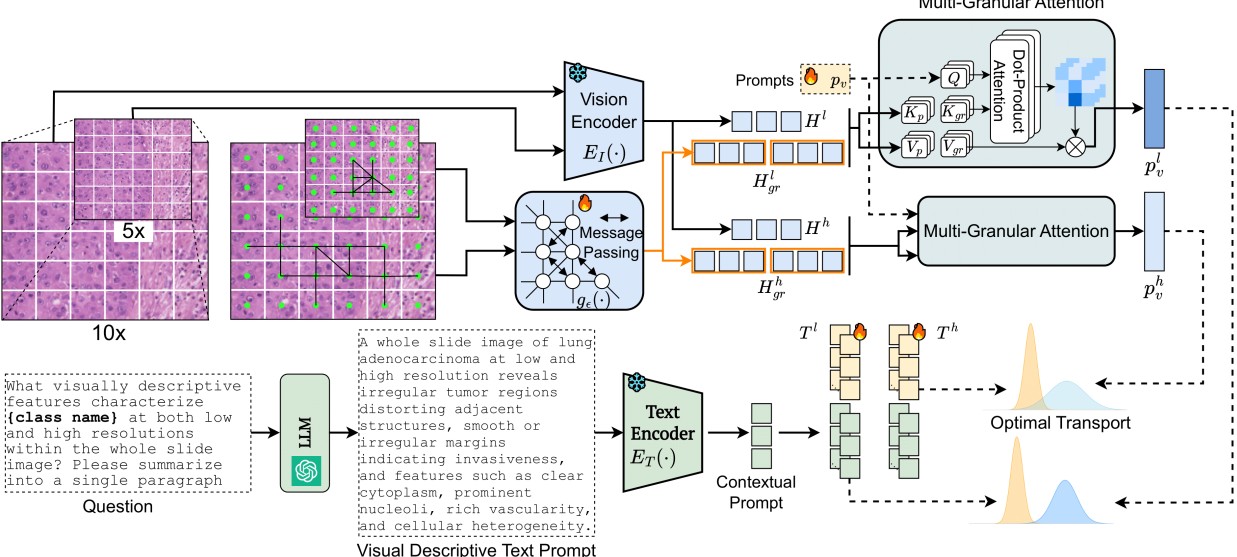

Figure 2: The pipeline of the proposed `MGPATH`. Low- and high-resolution image patches are processed with large language models to generate visual contextual descriptions (Section 3.2). Visual prompts are integrated with frozen features through multi-granular attention at both patch and group-of-patch levels 3.3. The final output is obtained by aligning visual and text embeddings using optimal transport (Section 3.4).

## 3.1 Bridging Pathology Visual and Text Encoders

To leverage `Prov-GigaPath`'s extensive pre-trained visual features for pathology, we implement lightweight adaptors that map image patch-level features to an embedding space aligned with the `PLIP` text encoder. These adaptors allow us to train joint image-text representations with parameter efficiency by updating only the adaptor weights.

Given a set of collected pathology image-text pairs $\{(\mathbf{I}_i, \mathbf{T}_i) | i = 1, 2.., N\}$ (Sec. 4), we denote by $E_I(.)$ be a pre-trained vision encoder from `Prov-GigaPath`, extracting patch-level feature, and $E_T(.)$ the pre-trained text encoder from `PLIP` model. Given a batch size of $B$ samples, the image and text embeddings are computed as $\mathbf{x}_i = E_I(\mathbf{I}_i) \in \mathbb{R}^{d_v}$, $\mathbf{t}_i = E_T(\mathbf{T}_i) \in \mathbb{R}^{d_t}$. *We then design two trainable adaptors $A_I(.)$ and $A_T(.)$, that maps* $(\mathbf{x}_i, \mathbf{t}_i)$ into the same hidden dimension $\mathbb{R}^d$ and minimizes the noise contrastive loss (Oord et al., 2018):

$$\mathcal{L}_{con} = \mathbb{E}_B \left[ -\log \frac{\exp\left(\cos(A_I(\mathbf{x}_i), A_T(\mathbf{t}_i))/\tau\right)}{\sum_j \exp\left(\cos(A_I(\mathbf{x}_i), A_T(\mathbf{t}_j))/\tau\right)} \right], \tag{1}$$

where cos(.) is the cosine similarity, and $\tau$ denotes for temperature of the softmax function. For parameter efficiency, we train only the adaptors $A_I(.), A_T(.)$ while keeping the `Prov-GigaPath` visual encoder and `PLIP` text encoder frozen. After optimizing equation 1, we use the outputs of the adaptors as visual and text embeddings for downstream tasks. Unless otherwise specified, we refer to this model as `MGPATH(PLIP-G)`.

## 3.2 Multi-Magnification Descriptive Text Prompts

To improve vision-language models (VLMs) for whole-slide image (WSI) analysis, designing effective text prompts is essential. Pathologists typically examine WSIs by first assessing tissue structures at low magnification before zooming in to analyze finer details such as nuclear size and shape. Inspired by this diagnostic workflow and the inherently multi-scale nature of WSIs, recent studies (Shi et al., 2024; Han et al., 2024) have introduced dual-scale visual descriptive text prompts to guide VLMs, leading to significant improvements in classification performance. Building on this observation, we further extend and refine this strategy to enhance model effectiveness.

First, to ensure that generated prompts remain robust across varying WSI magnifications, we design shared prompts that combine both high- and low-scale descriptive elements, treating them as contextual embeddings. Specifically, we leverage the API of a frozen language model (GPT-4) and query it with the prompt as Figure 3

> **LLM Prompt**
>
> ```
> What visually descriptive features characterize {class name} at both low and high resolutions
> within the whole-slide image?  Please summarize into a single paragraph.
> ```

Figure 3: LLM template prompt.

In the above query, we replace `{class name}` by specific categories, for e.g., they are invasive ductal carcinoma (IDC) and invasive lobular carcinoma (ILC) in the `TCGA-BRCA` dataset.

Second, at each low/high scale, rather than inserting *a single learnable text prompt* of length $K$ alongside a frozen contextual prompt from LLMs (Shi et al., 2024; Han et al., 2024), we propose *using $M$ learnable prompts*. This approach aims to capture different sub-regions or structural features within each patch that might be overlooked with only a single prompt. Specifically, we define visual descriptive text prompts for both low and high resolutions as follows:

$$
\begin{aligned}
\mathbf{T}_i^{(l)} &= \left\{ \left( [\omega_i^{(l)}]_1 \, [\omega_i^{(l)}]_2 \, ... [\omega_i^{(l)}]_K [\text{LLM context}] \right) |_{i=1}^M \right\} \\
\mathbf{T}_i^{(h)} &= \left\{ \left( [\omega_i^{(h)}]_{(1)} \, [\omega_i^{(h)}]_2 \, ... [\omega_i^{(h)}]_K [\text{LLM context}] \right) |_{i=1}^M \right\},
\end{aligned}
\tag{2}
$$

where $[\omega_i^{\beta}]_j, j \in [1, ..., K], i \in [1, .., M]$ are $KM$ *trainable textual prompts* for each resolution $\beta \in \{l, h\}$.

## 3.3 Granularity-aware Visual Prompt Learning

We propose to adapt visual prompts to frozen features extracted by a pre-trained vision encoder in the VLM model by taking into account the image patch level and spatial groupings of patches. Specifically, for each WSI $W$, we denote by $\left\{W^{(l)}, W^{(h)}\right\}$ are representations of $W$ at low and high magnification. We define a bag of multiple instances of $W$ as $I = \left\{I^{(l)}, I^{(h)}\right\}$ where $I^{(l)} \in \mathbb{R}^{N_l \times N_b \times N_b \times 3}$, $I^{(h)} \in \mathbb{R}^{N_h \times N_b \times N_b \times 3}$ with $N_l, N_h$ indicate the number of low and high-resolution image patches and $N_b$ is the patch size. Following prior works (Shi et al., 2024; Ilse et al., 2018; Lu et al., 2021; Han et al., 2024), we employ a non-overlapping sliding window technique to extract patches $I$ from the WSI.

### 3.3.1 Patches-based Prompting

The frozen image encoder $E_I(.)$ (or $A_I(E_I(.))$ in case of `MGPATH(PLIP-G)`) is used to map patches $I$ into a feature vector $H = \{H^{(l)} \in \mathbb{R}^{N_l \times d}, H^{(h)} \in \mathbb{R}^{N_h \times d}\}$ where $d$ denotes the feature dimension. To effectively consolidate the extensive set of patch features into a final slide-level representation, we introduce a set of learnable visual prompts $\mathbf{p}_v \in \mathbb{R}^{N_p \times d}$, which facilitate the progressive merging of patch features in $H^{(l)}$ (similarly for $H^{(h)}$) (Figure 2). In particular, we formulate $\mathbf{p}_v$ as Query and take all features in $H^{(l)}$ as the Keys $K_p^{(l)}$ and Values $V_p^{(l)}$ in in self-attention (Vaswani, 2017). We then associate $\mathbf{p}_v$ with patch features as:

$$
\mathbf{p}_{v,p}^{(l)} = \text{Normalize}\left( \text{SoftMax}\left( \frac{\mathbf{p}_v K_p^{(l)^T}}{\sqrt{d}} \right) V_p^{(l)} \right) + \mathbf{p}_v,
\tag{3}
$$

where Normalize(.) and Softmax(.) indicate the layer normalization operator and activation function respectively. Intuitively, equation 3 computes the correlations between the visual prompt $\mathbf{p}_v$ and all individual feature patches in $H^l$, subsequently grouping patches with high similarity to form fused prompt embeddings. However, since cancerous tissues in WSIs often appear as large, contiguous regions of adjacent image patches, this motivates the introduction of spatial patch group-based attention.

### 3.3.2 Spatial Patch Group-based Prompting

We build spatial correlations for multiple instances in $I$ by using their image patch coordinates inside each WSI $W$. In particular, taken $I^{(l)} = \left\{ I_1^{(l)}, I_2^{(l)}, ..., I_{N_l}^{(l)} \right\}$ with their corresponding extracted features in $H^{(l)} = \left\{ H_1^{(l)}, H_2^{(l)}, ..., H_{N_l}^{(l)} \right\}$, we construct a graph $G^{(l)} = (V^{(l)}, E^{(l)})$ to capture regional tissue structures where the set of vertices $V^{(l)} = I^{(l)}$, and $E^{(l)} \in \{0,1\}^{N_l \times N_l}$ is the set of edges. Edges in $E^{(l)}$ can be defined by linking inner patches to their K-nearest neighbors based on the coordinates. We define the node-feature embedding as $X^{(l)} = H^{(l)} \in \mathbb{R}^{N_l \times d}$ that associates each vertex $v_i^{(l)}$ with its feature node $x_i^{(l)} = H_i^{(l)}$.

We next design a trainable message-passing network $g_\epsilon(.)$ based on the graph attention layer (GAT) (Veličković et al., 2017) to capture the feature representation of each node and its local neighbors. The message passing of the GAT layer is formulated as:

$$\alpha_{i,j} = \frac{\exp\left(\sigma(a_s^T \Theta_s \, x_i^{(l)} + a_t^T \Theta_t \, x_j^{(l)})\right)}{\sum_{k \in \mathcal{N}(i) \cup \{i\}} \exp(\sigma(a_s^T \Theta_s \, x_i^{(l)} + a_t^T \Theta_t \, x_k^{(l)}))}$$
$$x_i^{(l)'} = \alpha_{i,i} \Theta_s \, x_i^{(l)} + \sum_{j \in \mathcal{N}(i)} \alpha_{i,j} \Theta_t \, x_j^{(l)}, \tag{4}$$

where $x_i^{(l)'}$ is aggregated features of $x_i^{(l)}$ with its local region after GAT layer, $\sigma(.)$ is the `LeakyReLU` activation function, $\mathcal{N}(i)$ denote the neighboring nodes of the $i$-th node, $\alpha_{i,j}$ are the attention coefficients and $a_s, a_t, \Theta_s, \Theta_t$ are weight parameters of $g_\epsilon(.)$.

After doing a message passing by $g_\epsilon(.)$, the graph of patch-image features $G^{(l)}$ is updated to $G^{(l)'}$, where each node now represents a super-node that encapsulates its corresponding feature region. We then squeeze all feature nodes in $G^{(l)'}$ as a vector $H_{gr}^{(l)}$ and treat them as another Keys $K_{gr}^{(l)}$ and Values $V_{gr}^{(l)}$ for region-level features. Similar to equation 3, we associate prompt $\mathbf{p}_v$ with those group-level features:

$$\mathbf{p}_{v,gr}^l = \text{Normalize}\left(\text{SoftMax}\left(\frac{\mathbf{p}_v K_{gr}^{(l)^T}}{\sqrt{d}}\right) V_{gr}^{(l)}\right) + \mathbf{p}_v. \tag{5}$$

The final output of our multi-granular is computed as:

$$\mathbf{p}_v^{(l)} = (1 - \alpha) \cdot \mathbf{p}_{v,p}^{(l)} + \alpha \cdot \mathbf{p}_{v,gr}^{(l)}, \tag{6}$$

which interpolates between image patches and spatial patch groups.

### 3.4 Optimal Transport for Visual-Text Alignment

Given descriptive text prompts $\mathbf{T}^{(l)}$ and $\mathbf{T}^{(h)}$ (equation 2) and visual prompt-guided slide features $\mathbf{p}_v^{(l)}$ and $\mathbf{p}_v^{(h)}$ (equation 6) for low and high resolutions, our goal is to maximize the similarity between slide and text embeddings for each class $c$. Rather than relying on cosine distance, as in prior works (Zhou et al., 2022b;a; Zhao et al., 2024; Qu et al., 2024a;a; Singh & Jaggi, 2020), we propose using optimal transport (OT)-based distance to capture a more nuanced cross-alignment between visual and text domains. Although OT has been explored for prompt learning in natural images and multi-modal learning (Kim et al., 2023; Chen et al., 2023; Nguyen et al., 2024a; Séjourné et al., 2023), we are the first to adapt it for whole-slide imaging (WSI), effectively handling the alignment of multi-magnification patches to capture rich structural details across scales.

**Recap OT:** Given two sets of points (features), we can represent the corresponding discrete distributions as follows:

$$\boldsymbol{\mu} = \sum_{i=1}^{M} p_i \delta_{f_i}, \quad \boldsymbol{\nu} = \sum_{j=1}^{N} q_j \delta_{g_j}, \tag{7}$$

where $\delta_f$ and $\delta_g$ represent Dirac delta functions centered at $\boldsymbol{f}$ and $\boldsymbol{g}$, respectively, and $M$ and $N$ indicate the dimensions of the empirical distribution. The weight vectors $\boldsymbol{p} = \{p_i\}_{i=1}^M$ and $\boldsymbol{q} = \{q_i\}_{j=1}^N$ lie within the $M$ and $N$-dimensional simplex, respectively, meaning they satisfy $\sum_{i=1}^M p_i = 1$ and $\sum_{j=1}^N q_j = 1$. The discrete optimal transport problem can then be expressed as:

$$\boldsymbol{T}^* = \underset{\boldsymbol{T} \in \mathbb{R}^{MXN}}{\arg\min} \sum_{i=1}^M \sum_{j=1}^N \boldsymbol{T}_{ij} \boldsymbol{C}_{ij}$$
$$\text{s.t.} \quad \boldsymbol{T}\boldsymbol{1}^N = \boldsymbol{\mu}, \quad \boldsymbol{T}^\top \boldsymbol{1}^M = \boldsymbol{\nu}. \tag{8}$$

where $\boldsymbol{T}^*$ is denoted as the optimal transport plan, which is optimized to minimize the total distance between the two probability vectors, $\boldsymbol{C}$ is the cost matrix which measures the distance between $\boldsymbol{f}_i$ and $\boldsymbol{g}_j$. We then define the OT distance between $\boldsymbol{\mu}$ and $\boldsymbol{\nu}$ as:

$$d_{\text{OT}}(\boldsymbol{\mu}, \boldsymbol{\nu}) = \langle \boldsymbol{T}^*, \boldsymbol{C} \rangle. \tag{9}$$

**Objective functions:** Given the visual prompt-guided slide features $\mathbf{p}_v^{(l)} \in \mathbb{R}^{N_p \times d}$ in equation 6 and the descriptive text prompts $\mathbf{T}^{(l)}$ in equation 2, we compute the textual embedding for $\mathbf{T}^{(l)}$ as $\mathbf{p}_t^{(l)} = E_T(\mathbf{T}^{(l)}) \in \mathbb{R}^{M \times d}$.

We next denote $\mathbf{T}_c^{(l)}$ as the input text prompts, $\left(\mathbf{p}_t^{(l)}\right)_c$ as the extracted textual embedding, and $\left(\mathbf{p}_v^{(l)}\right)_c$ as the visual prompt-guided slide features associated with class $c$. We then aim to minimize the distance between $\mathbf{T}_c^{(l)}$ and $\left(\mathbf{p}_v^{(l)}\right)_c$, indicated as $d_{\text{OT}}\left(\mathbf{T}_c^{(l)}, \left(\mathbf{p}_v^{(l)}\right)_c\right)$ in the paper, by computing optimal transport distance between $\left(\mathbf{p}_t^{(l)}\right)_c$ and $\left(\mathbf{p}_v^{(l)}\right)_c$. Specifically, we treat $\left(\mathbf{p}_t^{(l)}\right)_c \to \boldsymbol{F} = \{\boldsymbol{f}_i|_{i=1}^M\}$ and $\left(\mathbf{p}_v^{(l)}\right)_c \to \boldsymbol{G} = \left\{\boldsymbol{g}_j|_{j=1}^{N_p}\right\}$ and compute the cost matrix $\boldsymbol{C}$ as $\boldsymbol{C} = \left(\boldsymbol{1} - \boldsymbol{F}^T \boldsymbol{G}\right) \in \mathbb{R}^{M \times N_p}$, which used to compute $\boldsymbol{T}^*$ in equation 8 for estimate optimal transport distance defined in equation 9. Following the same procedure, we can also compute $d_{\text{OT}}\left(\mathbf{T}_c^{(h)}, \left(\mathbf{p}_v^{(h)}\right)_c\right)$ at high-resolution image patches. Then, the prediction probability is written as:

$$\mathrm{P}_c = \frac{\exp(2 - \sum_{k \in \{l,h\}} d_{\text{OT}}\left(\mathbf{T}_c^{(k)}, \left(\mathbf{p}_v^{(k)}\right)_c\right))}{\sum_{c'=1}^C \exp(2 - \sum_{k \in \{l,h\}} d_{\text{OT}}\left(\mathbf{T}_{c'}^{(k)}, \left(\mathbf{p}_v^{(k)}\right)_c\right))}, \tag{10}$$

where $\lambda_k$ controls contribution of each-resolution. Finally, we can train the model with the cross-entropy as:

$$\mathcal{L}_{class} = \text{Cross}(\mathrm{P}, \text{GT}), \tag{11}$$

with Cross(.) be the cross-entropy and GT denotes slide-level ground-truth.

The details for solvers of equation 9 and a relaxed version with unbalanced optimal transport are presented in Sections (D) and (D.1) in Appendix. Intuitively, using OT, in this case, offers several key advantages over cosine similarity. Pathology images exhibit complex, heterogeneous patterns that can be described from multiple perspectives. OT models these relationships as a distribution, enabling a more holistic alignment that handles variability and incomplete details while reducing noise from irrelevant prompts. This enhances the model's ability to generalize to unseen or complex disease cases.

## 4 Experiments

### 4.1 Settings

**Datasets for contrastive learning.** `PatchGastricADC22` (Tsuneki & Kanavati, 2022) consists of approximately 262K patches derived from WSI of H&E-stained gastric adenocarcinoma specimens, each paired with associated diagnostic captions collected from the University of Health and Welfare, Mita Hospital, Japan. `QUILT-1M` (Ikezogwo et al., 2024) includes approximately 653K images and one million pathology image-text pairs, gathered from 1,087 hours of educational histopathology videos presented by pathologists on YouTube. `ARCH` (Gamper & Rajpoot, 2021) is a pathology multiple-instance captioning dataset containing pathology

images at the bag and tile level. However, our work focuses on tile-level images from all datasets for our contrastive training strategy. In total, we collected approximately 923K images from these datasets.

**Downstream tasks.** For the classification task, the proposed method was evaluated in three datasets from the Cancer Genome Atlas Data Portal (The Cancer Genome Atlas (TCGA)): `TCGA-NSCLC`, `TCGA-RCC`, and `TCGA-BRCA`. We followed the `ViLa-MIL`(Shi et al., 2024) experimental settings for `TCGA-NSCLC` and `TCGA-RCC`, randomly selecting proportions for training, validation, and testing. For `TCGA-BRCA`, we adapted the training and testing slide ID from `MSCPT` (Han et al., 2024). The detailed description is included in the appendix section.

**Evaluation Metrics.** For all experiments, we report the Area Under the Curve (AUC), F1 score (F1), and accuracy (ACC) as evaluation metrics. Each experiment is repeated five times, and we present the mean and standard deviation across these runs to ensure robustness and reproducibility.

**Implementation Details.** We followed the `ViLa-MIL` preprocessing pipeline for tissue region selection and patch cropping. To integrate our attention module with `CLIP50` and `PLIP`, we extracted tile-level embeddings from their frozen vision encoders (1024-dimensional for `CLIP50` and 512 for `PLIP`). We used the visual encoder of `Prov-GigaPath` to produce 1536-dimensional embeddings. To align it with `PLIP`'s frozen text encoder, we developed two MLP-based adaptors that project both encoders into a shared feature space during a contrastive learning process, using datasets outlined in Section 4.

To implement spatial attention, we use a Graph Attention Network (GAT) to model spatial relationships between WSI patches. Each tile-level embedding serves as a node, connected to its left, right, top, and bottom neighbors, ensuring local spatial dependencies are captured. We then integrate spatial patch group-based attention $\mathbf{p}_{v,gr}$ into patch-based attention $\mathbf{p}_{v,p}$ using equation 6. The hyperparameter $\alpha$ (0 to 1) controls the balance between spatial context and prototype-based guidance.

In this paper, we leverage whole slide images at two resolution levels - low magnification (5x) and high magnification (10x) - to capture both tissue-level context and fine cellular detail. TThe whole slide images are processed by Otsu's algorithm to remove all the non-tissue regions. The cropped patch size is $256 \times 256$. All experiments were executed on a single NVIDIA A100 GPU node.

### 4.2 Comparison to State-of-the-Art

We compare our `MGPath` with state-of-the-art multi-instance learning methods, including `Maxpooling`, `Mean-pooling`, `ABMIL` (Ilse et al., 2018), `CLAM` (Lu et al., 2021), `TransMIL` (Shao

Table 1: Comparison of methods on `TCGA-BRCA` with 16-shot settings.

| | Methods | # Param. | TCGA-BRCA | | |
|---|---|---|---|---|---|
| | | | AUC | F1 | ACC |
| **CLIP ImageNet Pretrained** | Max-pooling | 197K | 60.42±4.35 | 56.40±3.58 | 68.55±6.54 |
| | Mean-pooling | 197K | 66.64±4.21 | 60.70±2.78 | 71.73±3.59 |
| | ABMIL (Ilse et al., 2018) | 461K | 69.24±3.90 | 61.72±3.36 | 72.77±3.15 |
| | CLAM-SB (Lu et al., 2021) | 660K | 67.80±5.14 | 60.51±5.01 | 72.46±4.36 |
| | CLAM-MB (Lu et al., 2021) | 660K | 60.81±4.87 | 55.48±4.96 | 67.31±4.19 |
| | TransMIL (Shao et al., 2021) | 2.54M | 65.62±3.20 | 60.75±4.04 | 67.52±4.16 |
| | DSMIL (Li et al., 2021) | 462K | 66.18±3.08 | 59.35±3.18 | 67.52±1.56 |
| | RRT-MIL (Tang et al., 2024) | 2.63M | 66.33±4.30 | 61.14±5.93 | 71.21±8.94 |
| | CoOp (Zhou et al., 2022b) | 337K | 68.86±4.35 | 61.64±2.40 | 71.08±3.22 |
| | CoCoOp (Zhou et al., 2022a) | 370K | 69.13±4.27 | 61.48±2.62 | 72.41±1.87 |
| | Metaprompt (Zhao et al., 2024) | 360K | 69.12±4.46 | 63.39±4.38 | 74.65±7.20 |
| | TOP (Qu et al., 2024a) | 2.11M | 69.74±3.14 | 63.39±4.62 | 74.41±5.27 |
| | ViLa-MIL (Shi et al., 2024) | 2.77M | 72.25±6.16 | 62.04±2.38 | 75.01±6.14 |
| | MSCPT (Han et al., 2024) | 1.35M | 74.56±4.54 | **65.59±1.85** | 75.82±2.38 |
| | **MGPATH (ViT)** | 592K | **74.96±6.98** | 64.60±5.39 | **77.10±2.39** |
| **FVM** | CONCH (Lu et al., 2024) | 110M | 84.11±15.44 | 65.63±10.81 | 73.24±8.89 |
| | QUILT (Ikezogwo et al., 2024) | 63M | 73.48±10.57 | 63.78±8.72 | 73.26±10.13 |
| **PLIP Pathology Pretrained** | Max-pooling | 197K | 66.50±2.74 | 61.50±2.88 | 71.57±4.82 |
| | Mean-pooling | 197K | 71.62±2.41 | 66.34±2.96 | 74.45±2.49 |
| | ABMIL (Ilse et al., 2018) | 461K | 72.41±4.25 | 63.04±3.62 | 74.09±4.38 |
| | CLAM-SB (Lu et al., 2021) | 660K | 72.34±6.17 | 65.51±3.28 | 76.16±4.36 |
| | CLAM-MB (Lu et al., 2021) | 660K | 73.41±3.76 | 66.11±1.94 | 77.88±2.30 |
| | TransMIL (Shao et al., 2021) | 2.54M | 74.98±6.01 | 67.50±6.00 | 77.04±6.14 |
| | DSMIL (Li et al., 2021) | 462K | 71.44±2.72 | 64.48±1.64 | 75.26±2.28 |
| | RRT-MIL (Tang et al., 2024) | 2.63M | 71.21±6.46 | 64.15±1.38 | 75.92±5.10 |
| | CoOp (Zhou et al., 2022b) | 337K | 71.53±2.45 | 64.84±2.40 | 74.22±5.02 |
| | CoCoOp (Zhou et al., 2022a) | 370K | 72.65±4.63 | 66.63±3.55 | 66.98±3.35 |
| | Metaprompt (Zhao et al., 2024) | 360K | 74.86±4.25 | 65.03±1.81 | 77.88±3.22 |
| | TOP (Qu et al., 2024a) | 2.11M | 76.13±6.01 | 66.55±1.72 | 78.58±5.30 |
| | ViLa-MIL (Shi et al., 2024) | 2.77M | 74.06±4.62 | 66.03±1.81 | 78.12±4.88 |
| | MSCPT (Han et al., 2024) | 1.35M | 75.55±5.25 | 67.46±2.43 | 79.14±2.63 |
| | **MGPATH (PLIP)** | 592K | 79.02±6.43 | 68.25±4.42 | **79.65±1.72** |
| | **MGPATH (PLIP-G)** | 5.35M | **87.36±1.85** | **73.13±3.49** | 79.56±4.77 |

et al., 2021), `DSMIL` (Li et al., 2021), `GTMIL` (Zheng et al., 2022), `DTMIL` (Zhang et al., 2022a), `RRT-MIL` (Tang et al., 2024) and `IBMIL` (Lin et al., 2023), and vision-language methods, including `CoOp` (Zhou et al., 2022b), `CoCoOp` (Zhou et al., 2022a), `Metaprompt` (Zhao et al., 2024), `TOP` (Qu et al., 2024a), `ViLa-MIL` (Shi et al., 2024), `MSCPT` (Han et al., 2024), `QUILT` (Ikezogwo et al., 2024), `CONCH` (Lu et al., 2024). Among these, `QUILT` and `CONCH` are foundation vision-language models (FVM).

Table 2: Comparison of methods on `TCGA-NSCLC`, and `TCGA-RCC` datasets with 16-shots settings.

| Methods | # Param. | TCGA-NSCLC | | | TCGA-RCC | | |
|---|---|---|---|---|---|---|---|
| | | AUC | F1 | ACC | AUC | F1 | ACC |
| Max-pooling | 197K | 53.0±6.0 | 45.8±8.9 | 53.3±3.4 | 67.4±4.9 | 46.7±11.6 | 54.1±4.8 |
| Mean-pooling | 197K | 67.4±7.2 | 61.1±5.5 | 61.9±5.5 | 83.3±6.0 | 60.9±8.5 | 62.3±7.4 |
| ABMIL (Ilse et al., 2018) | 461K | 60.5±15.9 | 56.8±11.8 | 61.2±6.1 | 83.6±3.1 | 64.4±4.2 | 65.7±4.7 |
| CLAM-SB (Lu et al., 2021) | 660K | 66.7±13.6 | 59.9±13.8 | 64.0±7.7 | 90.1±2.2 | 75.3±7.4 | 77.6±7.0 |
| CLAM-MB (Lu et al., 2021) | 660K | 68.8±12.5 | 60.3±11.1 | 63.0±9.3 | 90.9±4.1 | 76.2±4.4 | 78.6±4.9 |
| TransMIL (Shao et al., 2021) | 2.54M | 64.2±8.5 | 57.5±6.4 | 59.7±5.4 | 89.4±5.6 | 73.0±7.8 | 75.3±7.2 |
| DSMIL (Li et al., 2021) | 462K | 67.9±8.0 | 61.0±7.0 | 61.3±7.0 | 87.6±4.5 | 71.5±6.6 | 72.8±6.4 |
| GTMIL (Zheng et al., 2022) | N/A | 66.0±15.3 | 61.1±12.3 | 63.8±9.9 | 81.1±13.3 | 71.1±15.7 | 76.1±12.9 |
| DTMIL (Zhang et al., 2022a) | 986.7K | 67.5±10.3 | 57.3±11.3 | 66.6±7.5 | 90.0±4.6 | 74.4±5.3 | 76.8±5.2 |
| IBMIL (Lin et al., 2023) | N/A | 69.2±7.4 | 57.4±8.3 | 66.9±6.5 | 90.5±4.1 | 75.1±5.2 | 77.2±4.2 |
| ViLa-MIL (Shi et al., 2024) | 8.8M/47M | 74.7±3.5 | 67.0±4.9 | 67.7±4.4 | 92.6±3.0 | 78.3±6.9 | 80.3±6.2 |
| CONCH ((Lu et al., 2024)) | 110M | 89.46±10.2 | 78.5±9.31 | 78.78±9.1 | 88.08±4.59 | 78.21±4.2 | 71.67±19.4 |
| QUILT (Ikezogwo et al., 2024) | 63M | 79.66±13.19 | 72.30±13.35 | 72.42±13.24 | 96.92±1.6 | 78.46±5.55 | 86.34±1.56 |
| **MGPath (CLIP)** | 1.6M/39M | 77.2±1.3 | 70.9±2.0 | 71.0±2.1 | 92.1 ± 2.8 | 76.5 ± 5.2 | 81.7 ± 2.9 |
| **MGPath (PLIP)** | 592K | 83.6 ± 4.5 | 76.41 ± 4.8 | 76.5 ± 4.8 | 94.7 ± 1.6 | 78.6 ± 4.9 | 83.6 ± 3.5 |
| **MGPath (PLIP-G)** | 5.35M | **93.02±2.99** | **84.64±4.75** | **84.77±4.67** | **98.2±0.31** | **88.33±3.41** | **91.72±1.74** |

### 4.3 MGPATH versions

We release four `MGPATH` variants. The ResNet-50 backbone CLIP-based models (`MGPATH(CLIP)`) for `TCGA-NSCLC` and `TCGA-RCC`, and a ViT-B/16 backbone (`MGPATH(ViT)`) for `TCGA-BRCA`. A pure `PLIP` version (`MGPATH(PLIP)`) is also provided. Finally, `MGPATH(PLIP-G)` configuration-combining the PLIP text encoder with the vision encoder from `Prov-GigaPath` pretrained on large-scale pathology dataset-is employed for all datasets.

### 4.4 Results on Few-shot and Zero-shot Settings.

**MGPATH with CLIP and PLIP backbones outperform several competitive MIL and VLM methods.** As shown in Tables 2 and 1, our `MGPATH(CLIP)`, `MGPATH(ViT)`, and `MGPATH(PLIP)` outperforms several baseline models and achieves significant improvements over other VLMs with similar architectures, such as ViLa and MSCPT, under 16-shots setting. The performance gain is particularly notable with the `PLIP` backbone. For example, on `TCGA-BRCA`, `MGPATH(ViT)` achieves an accuracy of 77.10%, compared to 75.82% for MSCPT and 75.01% for ViLA-MIL. Additionally, with the `PLIP` backbone, `MGPATH(PLIP)` surpasses `MSCPT` and `ViLa-MIL` by margins of approximately 3.5% to 5%.

**MGPATH(PLIP-G) is a strong pre-trained VLM.** By incorporating pathology-specific features from `Prov-GigaPath` (Xu et al., 2024), which is pre-trained on 1.3 billion pathology images, and `PLIP` text encoder, `MGPATH (PLIP-G)` model demonstrated strong performance across multiple metrics on the `TCGA-NSCLC`, `TCGA-RCC`, and `TCGA-BRCA` datasets. When compared to other foundation VLMs such as `CONCH` and `QUILT`, our model consistently outperforms them. For example, we achieve a 3% improvement in AUC over `CONCH` on both the `TCGA-BRCA` and `TCGA-NSCLC` datasets.

**MGPATH(PLIP-G) achieves competitive performance in zero-shot tasks.** We evaluate the zero-shot capabilities of our model on three datasets and compare its performance against foundation VLMs such as CONCH, QUILT, and PLIP. The results, summarized in Table 3, show that the proposed VLM model achieves the best average performance across datasets, followed by CONCH and PLIP. This consistent top-tier performance across multiple benchmarks underscores the robustness and generalizability of our model.

**Evaluating MGPATH(PLIP-G) with fewer Samples.** To assess the impact of number of training samples (number of shots $K$) on `MGPATH(PLIP-G)`, we conducted experiments under 2-, 4-, 8-, 16-shots on `TCGA-BRCA`. Figure 4 presents box plots - each summaries five independent runs - that track F1 scores. Overall, `MGPATH(PLIP-G)` consistently surpassed the baselines across 2, 4, 8, and 16 shots.

Table 3: Zero-shot classification performance on TCGA-NSCLC, TCGA-RCC, and TCGA-BRCA datasets. Metrics reported include balanced accuracy (B-Acc) and weighted F1-score (W-F1).

| Zero-shot | TCGA-NSCLC | | TCGA-RCC | | TCGA-BRCA | | Average | |
|---|---|---|---|---|---|---|---|---|
| | B-Acc | W-F1 | B-Acc | W-F1 | B-Acc | W-F1 | B-Acc | W-F1 |
| QuiltNet | 61.3 | 56.1 | 59.1 | 51.8 | 51.3 | 40.1 | 57.23 | 49.33 |
| CONCH | **80.0** | **79.8** | 72.9 | 69.1 | 64.0 | 61.2 | 72.3 | 70.03 |
| PLIP | 70.0 | 68.5 | 50.7 | 46.0 | 64.7 | 63.8 | 61.8 | 59.43 |
| PLIP-G (Our) | 72.7 | 72.6 | **81.3** | **81.4** | **70.0** | **69.9** | **74.67** | **74.63** |

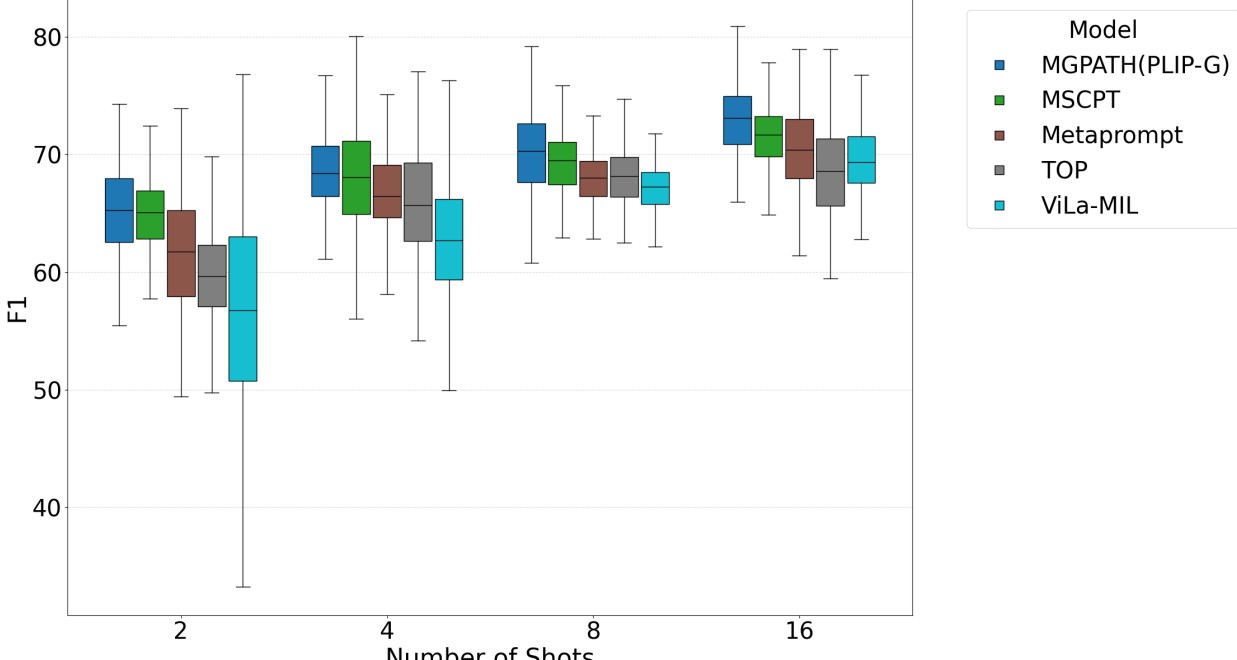

Figure 4: F1 scores of `MGPATH(PLIP-G)` under 2-, 4-, 8-, and 16-shots training on `TCGA-BRCA`.

## 4.5 Ablation Studies

**Multi-Granular Prompt Learning.** In Table 4, we show the performance of `MGPATH(CLIP)` and `MGPATH(PLIP-G)` with and without multi-granular on two datasets `TCGA-NSCLC` and `TCGA-RCC`. It shows that using `M-Gran` improves the final performance of `MGPATH(CLIP)` and `MGPATH(PLIP-G)` on both datasets. We further investigate the impact of ratio when combining prototype-guided attention with graph spatial attention on `TCGA-NSCLC`. Table 5 shows that with a ratio of 0.2/0.8 (0.2 for spatial attention obtained from graph structure and 0.8 for prototype-guided attention), `MGPATH(CLIP)` achieves the highest performance.

**PLIP enhanced Prov-GigaPath.** We validate the use of `Prov-GigaPath` and `PLIP` under the following settings: (i) using full vision-language PLIP model; (ii) combining `Prov-GigaPath` with `PLIP` through the MLP layer which was pre-trained on the large-scale dataset; (iii) integrating `Prov-GigaPath` with `PLIP` through adaptor layers which were randomly initialized; (iv) utilizing `Prov-GigaPath` and an adaptor layer to map to the class output and only train MLP and last FFN layer of slide encoder; (v) only using `Prov-GigaPath` and an MLP layer to map to the class output and train MLP, the query matrix of last layer and the last FFN layer of slide encoder. Table 6 shows that using `Prov-GigaPath` combined with `PLIP` (`MGPATH(PLIP-G)`) boosts the final performance compared to only using `PLIP` (`MGPATH(PLIP)`) or `Prov-GigaPath`.

**OT as Alignment between Contextual Prompts.** Table 7 validates the use of OT in our `MGPATH(CLIP)` on `TCGA-NSCLC` and `TCGA-RCC`. We see that using OT helps to boost the performance of `MGPATH(CLIP)`

compared to the use of cosine. It also shows that the number of prompt vectors depends on each dataset. In the appendix, we also run with another version using unbalanced optimal transport (UoT). We observe that both UoT and OT provide good alignment quality, with UoT slightly outperforming OT. However, this advantage comes at the cost of increased running time.

Table 4: Ablation studies on multi-granular (M-Gran) on the proposed models' performance.

| Configurations | TCGA-NSCLC | | |
|---|---|---|---|
| | AUC | F1 | ACC |
| MGPATH (CLIP) | 77.2±1.3 | 70.9±2.0 | 71.0±2.1 |
| - w/o M-Gran (CLIP) | 74.6±2.2 | 67.8±2.4 | 67.8±2.5 |
| MGPATH (PLIP-G) | 91.7±3.6 | 84.2±4.6 | 84.4±4.5 |
| - w/o M-Gran (PLIP-G) | 90.6±4.5 | 82.4±5.7 | 82.5±5.7 |
| | TCGA-RCC | | |
| MGPATH (CLIP) | 92.1±2.8 | 76.5±5.2 | 81.7±2.9 |
| - w/o M-Gran (CLIP) | 91.6±3.5 | 72.3±6.4 | 80.2±4.4 |
| MGPATH (PLIP-G) | 98.1±0.6 | 85.7±1.1 | 89.9±2.0 |
| - w/o M-Gran (PLIP-G) | 98.1±0.6 | 85.0±4.0 | 89.3±3.0 |

Table 6: Ablation studies on adaptor learning for Prov-GiGa-Path and PLIP. PLIP-G denotes for mixed version between Prov-GigaPath and PLIP.

| Methods | # Param. | TCGA-NSCLC | | |
|---|---|---|---|---|
| | | AUC | F1 | ACC |
| MGPATH (PLIP) | 592K | 83.6±4.5 | 76.41±4.8 | 76.5±4.8 |
| MGPATH (PLIP-G) | 5.35M | 93.02±2.99 | 84.64±4.75 | 84.77±4.67 |
| MGPATH(PLIP-G) Random Adaptors | 5.35M | 91.4±4.2 | 82.8±5.7 | 83.0±5.6 |
| Prov-GigaPath Tuning (MLP + last FFN) | 4.7M | 62.7±3.5 | 64.66±5.3 | 52.8±3.4 |
| Prov-GigaPath Tuning (MLP + last Q-ViT) | 5.8M | 83.1±6.9 | 74.3±7.5 | 75.8±6.1 |

Table 7: Contribution of OT and multiple descriptive text prompts.

Table 5: Ablation studies investigate the effect of the ratio combines spatial attention ($\alpha$) and prototype-guided attention in `MGPATH(CLIP)`.

| Configurations | TCGA-NSCLC | | |
|---|---|---|---|
| | AUC | F1 | ACC |
| $\alpha = 0.2$ | 77.2±1.3 | 70.9±2.0 | 71.0±2.1 |
| $\alpha = 0.5$ | 73.7±3.1 | 67.4±2.6 | 67.8±2.7 |
| $\alpha = 0.8$ | 72.2±5.2 | 66.4±5.5 | 66.8±5.2 |

| Methods | TCGA-NSCLC | | |
|---|---|---|---|
| | AUC | F1 | ACC |
| MGPATH (OT, 4 text prompts) | 77.2±1.3 | 70.9±2.0 | 71.0±2.1 |
| MGPATH (OT, 2 text prompts) | 77.2±1.3 | 70.9±2.0 | 71.0±2.1 |
| MGPATH (Cosine, 2 text prompts) | 75.8±3.7 | 68.3±4.5 | 68.4±4.5 |
| | TCGA-RCC | | |
| MGPATH (OT, 4 text prompts) | 92.1±2.8 | 76.5±5.2 | 81.7±2.9 |
| MGPATH (OT, 2 text prompts) | 92.1±2.6 | 75.6±3.9 | 80.4±2.4 |
| MGPATH (Cosine, 4 text prompts) | 91.8±2.8 | 75.9±4.3 | 80.5±2.6 |

**Effect of Large Language Models.** To further validate the impact of different LLMs on `MGPATH(PLIP-G)`'s performance, we conducted experiments with text prompts generated by various LLMs, including `GPT-4o` (OpenAI, 2024), `Grok-3` (xAI, 2025), `GPT-o3` (OpenAI, 2025), `Gemini-2.5-pro` (Team et al., 2023), `Deepseek-R1` (DeepSeek-AI, 2025), and `Mistral Medium 3` (Mistral AI, 2024). The results in 16-shot settings on `TCGA-BRCA` are represented in Table 8. The experimental results demonstrated that `MGPATH(PLIP-G)` consistently exceeded the baseline across all text prompt sources. The model achieved the best performance with `Mistral Medium 3` (Mistral AI, 2024), which produced the highest AUC (93.95 ± 2.71), F1 score (88.53 ± 4.29), and accuracy (88.63 ± 4.26). This experiment demonstrated the robustness of the performance of the proposed method in various LLMs and highlighted the critical role of precise visual descriptions on model performance enhancement.

Table 8: Ablation experiments of input text prompts from different LLM

| | TCGA-BRCA | | |
|---|---|---|---|
| Methods | AUC | F1 | ACC |
| GPT-4o (OpenAI, 2024) | 93.02 ± 2.99 | 84.64 ± 4.75 | 84.77 ± 4.67 |
| Grok-3 (xAI, 2025) | 93.07 ± 3.56 | 85.50 ± 4.53 | 85.58 ± 4.51 |
| GPT-o3 (OpenAI, 2025) | 93.87 ± 3.31 | 86.19 ± 5.28 | 86.29 ± 5.26 |
| Gemini-2.5-pro (Team et al., 2023) | 93.17 ± 3.12 | 86.72 ± 3.50 | 86.80 ± 3.48 |
| Deepseek-R1 (DeepSeek-AI, 2025) | 93.40 ± 2.80 | 86.87 ± 4.68 | 87.01 ± 4.63 |
| **Mistral Medium 3** (Mistral AI, 2024) | **93.95 ± 2.71** | **88.53 ± 4.29** | **88.63 ± 4.26** |

### 4.6 Discussion

While we demonstrate significant improvements in few-shot and zero-shot WSI classification across several settings, this paper does not explore other important challenges. For example, how can we scale the current attention mechanism to handle even larger image patches (e.g., using Flash Attention (Dao et al., 2022)), or extend the model from classification to tumor segmentation tasks (Khened et al., 2021). Additionally, the potential for extending GiGaPath to integrate with other large-scale VLM models, such as `CONCH` (Lu et al., 2024), remains unexplored.

## 5 Conclusion

High-resolution WSI is crucial for cancer diagnosis and treatment but presents challenges in data analysis. Recent VLM approaches, which utilize few-shot and weakly supervised learning, have shown promise in handling complex whole-slide pathology images with limited annotations. However, many overlook the hierarchical relationships between visual and textual embeddings, ignoring the connections between global and local pathological details or relying on non-pathology-specific pre-trained models like `CLIP`. Additionally, previous metrics lack precision in capturing fine-grained alignments between image-text pairs. To address these gaps, (i) we propose `MGPATH(PLIP-G)`, which integrates `Prov-GigaPath` with `PLIP`, cross-aligning them with 923K domain-specific image-text pairs. (ii) Our multi-granular prompt learning approach captures hierarchical tissue details effectively, (iii) while OT-based visual-text distance ensures robustness against data augmentation perturbations. Extensive experiments on three cancer subtyping datasets demonstrate that `MGPATH` and its variants achieve state-of-the-art results in WSI classification. We expect that this work will pave the way for combining large-scale domain-specific models with multi-granular prompt learning and optimal transport to enhance few-shot learning in pathology.

## 6 Broader Impact Statement

`MGPATH` is developed to investigate the feasibility of applying vision-language models (VLMs) to pathology under few-shot learning settings, simulating scenarios with extremely limited training samples, such as rare diseases. However, the model is not intended for clinical use as a diagnostic tool or decision support system. It is not designed to replace professional medical judgment or diagnosis.

## 7 Acknowledgement

This work is supported in part by funds from the German Ministry of Education and Research (BMBF) under grant agreements *No. 01D2208A* and *No. 01KD2414A* (project FAIrPaCT). The authors gratefully acknowledge the computing time granted by the KISSKI project. The calculations for this research were conducted with computing resources under the project *kisski-umg-fairpact-2*. The authors also acknowledge the computing time granted by the Resource Allocation Board and provided on the supercomputer Emmy/Grete at NHR-Nord@Göttingen as part of the NHR infrastructure. The calculations for this research were conducted with computing resources under the project *nim00014*.

The project "Development of an intelligent collaboration service for AI-based collaboration between rescue services and central emergency rooms" (acronym: CONNECT_ED) is funded by the German Federal Ministry of Education and Research under grant number *16SV8977* and by the joint project *KISSKI* under grant number *1IS22093E*.

The authors thank the International Max Planck Research School for Intelligent Systems (IMPRS-IS) for supporting Duy M. H. Nguyen. Duy M. H. Nguyen and Daniel Sonntag are also supported by the XAINES project (BMBF, *01IW20005*), No-IDLE project (BMBF, *01IW23002*), and the Endowed Chair of Applied Artificial Intelligence, Oldenburg University. Anh-Tien Nguyen was a member of the Ph.D. program "Genome Science" – International Max Planck Research School.

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

## Appendix

## A    Description of Dataset Splitting

`TCGA-BRCA`. This dataset contains 1056 whole slide images of breast invasive carcinoma. To conduct fair experiments, we adapted training and testing slides provided by the GitHub repository of `MSCPT` (Han et al., 2024). In the `MSCPT` setup, 20% of the dataset was allocated for training, while the remaining 80% (833 slides) served as the test set. A fixed set of 16-shot WSIs was randomly sampled from the training set. Additionally, `MSCPT` specified the exact training and testing slides used in its experiments. However, there are 35 slides in which we got errors in the pre-processing steps; thus, we replaced those slides with the other ones (same number of WSI per class) downloaded from Cancer Genome Atlas (TCGA) Data Portal (GDC) (The Cancer Genome Atlas (TCGA)).

`TCGA-RCC` & `TCGA-NSCLC`. We adopt the same data splitting as in `ViLa-MIL` (Shi et al., 2024), using 16-shot samples for training in each dataset. For testing, 192 samples were used for `TCGA-RCC` and 197 samples were used for `TCGA-NSCLC`.

### A.1    Other hyper-parameters

For all experiments, we trained `MGPATH` with the Adam optimizer with a learning rate of $9 \times 10^{-6}$ and a weight decay of $1 \times 10^{-5}$ to fine-tune all versions of `MGPATH` presented in Tables  1 and  2. The training process was conducted for a maximum of 200 epochs, with a batch size set to 1. The best checkpoints are picked based validation performance with F1 score.

### A.2    Baseline Setups

`TCGA-BRCA`: The baselines in **Table  1** are sourced from the MSCPT (Han et al., 2024) paper, where various methods are evaluated using two backbones: `Vision Transformer (ViT)` (Alexey, 2020) from the `CLIP` model (top section of Table  1) and `PLIP` (Huang et al., 2023) (bottom section of Table  1). In this context, we introduce three variations of `MGPATH(ViT)`, `MGPATH(PLIP)`, and `MGPATH(PLIP-G)`, where all versions utilize frozen vision and text encoders.

`TCGA-RCC` & `TCGA-NSCLC`: The baselines in **Table  2** are adapted from `ViLa-MIL` (Shi et al., 2024) where methods employ ResNet-50 from the CLIP model as the primary backbone. We present the results using three architectures: `MGPATH(CLIP)`, `MGPATH(PLIP)`, and `MGPATH(PLIP-G)`. With `ResNet-50`, we follow the `ViLa-MIL` approach by training the text encoder and reporting performance for this setup. To assess efficiency, we provide the total parameter counts for both `ViLa-MIL` and `MGPATH`, considering scenarios with frozen backbones and trainable text encoders. For `MGPATH(PLIP)` and `MGPATH(PLIP-G)`, all visual and text encoders are kept frozen.

`CONCH` & `QUILT`: We download the pre-trained weights of these foundation models and adapt them for zero-shot evaluation on TCGA datasets following the authors' guidelines from  (Lu et al., 2024), which randomly sample 75 samples for each class. For few-shot settings, since official implementations are not provided, we initialize the models with their pre-trained weights and allow fully fine-tune the text encoder and evaluate on the same subsets that we use for other baselines. While `CONCH` provides prompts for the datasets in its publication, `QUILT` does not. Therefore, we fine-tune the model using `CONCH`'s prompts and our own generated prompts for `QUILT`.

## B    Impact of PLIP enhanced Prov-GigaPath

Figure 5 presents the AUC curves for three randomly selected folds, illustrating the impact of `Prov-GigaPath` on model performance. The results show that integrating `Prov-GigaPath` leads to consistently higher AUC values across all folds, demonstrating its effectiveness in enhancing the proposed model. Notably, the improvements are most pronounced during the early training epochs, where the model converges faster and achieves more stable performance compared to the baseline. This suggests that `Prov-GigaPath` facilitates better feature extraction and generalization, ultimately leading to a more robust model.

Table 9: Comparison of message passing algorithms in `MGPATH`, including `GAT-CONV`, `Graph Isomorphism Network (GIN)`, and `Graph Convolutional Network (GCN)`. Performance is evaluated on the `TCGA-NSCLC` dataset using 5-fold cross-validation.

Figure 5: AUC performance comparison over epochs for `PLIP` (blue) `MGPATH(PLIP-G)` (red). `MGPATH(PLIP-G)` achieved achieving more stable and higher values, particularly in the early epochs.

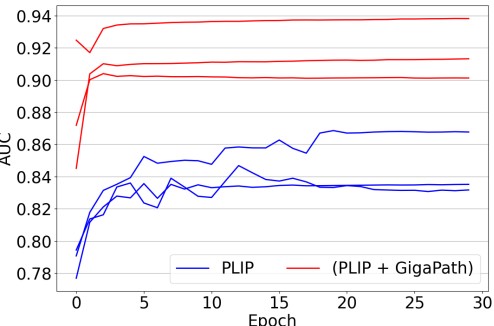

| Configurations | TCGA-NSCLC | | |
|---|---|---|---|
| | AUC | F1 | ACC |
| MGPATH (GAT CONV) | 77.2±1.3 | 70.9±2.0 | 71.0±2.1 |
| MGPATH (GIN) | 77.1±2.9 | 69.8±3.9 | 69.9±4.0 |
| MGPATH (GCN) | 75.1±2.9 | 67.6±2.5 | 67.1±2.8 |

## C  Ablation Study on Message Passing Networks

In Table 9, we evaluate the performance of `MGPATH(CLIP)` using the `Graph Attention Network (GAT-CONV)` against alternatives like the `Graph Isomorphism Network (GIN)` (Xu et al., 2019b) and the `Graph Convolutional Network (GCN)` (Kipf & Welling, 2017). The results show that `MGPATH (GIN)` achieves comparable performance to `MGPATH(CLIP)`(GAT-CONV), however, with higher variance. In contrast, `MGPATH(CLIP)`(GAT-CONV) significantly outperforms the `GCN`-based version, likely due to `GAT`'s ability to dynamically assign attention weights to neighboring image patches, enabling it to prioritize the most relevant neighbors for each node.

## D  Additional Details on Optimal Transport Distance

The following paragraphs will provide detailed information on the implementation of (un-balanced) optimal transport (OT) (Villani et al., 2009; Peyré et al., 2019) and specifically the alignment of prompt-guided visual-text distances in `MGPATH`.

### D.1  OT Formulation and Efficient Solver

Given two set of feature embeddings $\boldsymbol{F} = \left\{\boldsymbol{f}_i|_{i=1}^M\right\} \in \mathbb{R}^{M \times d}$ and $\boldsymbol{G} = \left\{\boldsymbol{g}_j|_{j=1}^N\right\} \in \mathbb{R}^{N \times d}$, we can represent them as two discrete distributions $\boldsymbol{\mu}$ and $\boldsymbol{\nu}$ by:

$$\boldsymbol{\mu} = \sum_{i=1}^M p_i \delta_{\boldsymbol{f}_i}, \quad \boldsymbol{\nu} = \sum_{j=1}^N q_j \delta_{\boldsymbol{g}_j}, \tag{12}$$

where $\delta_{\boldsymbol{f}_i}$ and $\delta_{\boldsymbol{g}_j}$ represent Dirac delta functions centered at $\boldsymbol{F}$ and $\boldsymbol{G}$, respectively and the weights are elements of the marginal $\boldsymbol{p} = \{p_i\}_{i=1}^M$ and $\boldsymbol{q} = \{q_i\}_{j=1}^N$ and can be selected as the uniform weight with $\sum_{i=1}^M p_i = 1, \sum_{j=1}^N q_j = 1$.

Then we can compute the distance between $\boldsymbol{F}$ and $\boldsymbol{G}$ through $\boldsymbol{\mu}$ and $\boldsymbol{\nu}$ (Eq.(9)) as

$$d_{\mathrm{OT}}(\boldsymbol{\mu}, \boldsymbol{\nu}) = \langle \boldsymbol{T}^*, \boldsymbol{C} \rangle. \tag{13}$$

where

$$\boldsymbol{T}^* = \underset{\boldsymbol{T} \in \mathbb{R}^{M \times N}}{\arg \min} \sum_{i=1}^M \sum_{j=1}^N \boldsymbol{T}_{ij} \boldsymbol{C}_{ij}$$

$$\text{s.t.} \quad \boldsymbol{T}\mathbf{1}^N = \boldsymbol{\mu}, \quad \boldsymbol{T}^\top \mathbf{1}^M = \boldsymbol{\nu}. \tag{14}$$

with $\boldsymbol{C} \in \mathbb{R}^{M \times N}$ is the cost matrix which measures the distance between $\boldsymbol{f}_i \in \boldsymbol{\mu}$ and $\boldsymbol{g}_j \in \boldsymbol{\nu}$.

Because directly solving Eq equation 14 is high-computational costs ($O(n^3 \log n)$ with $n$ proportional to $M$ and $N$), Sinkhorn algorithm (Cuturi, 2013) is proposed to approximate solution by solving a regularized problem:

$$\boldsymbol{T}^* = \underset{\boldsymbol{T} \in \mathbb{R}^{MXN}}{\arg\min} \sum_{i=1}^{M}\sum_{j=1}^{N} \boldsymbol{T}_{ij}\boldsymbol{C}_{ij} - \lambda H(\boldsymbol{T})$$

$$\text{s.t.} \quad \boldsymbol{T}\boldsymbol{1}^N = \boldsymbol{\mu}, \quad \boldsymbol{T}^\top \boldsymbol{1}^M = \boldsymbol{\nu}. \tag{15}$$

where $H(\boldsymbol{T}) = \sum_{ij}\boldsymbol{T}_{ij}\log\boldsymbol{T}_{ij}$ be an entropy function and $\lambda > 0$ is the regularization parameter. The optimization problem in Eq. equation 15 is strictly convex, allowing us to achieve a solution efficiently with fewer iterations as outlined below:

$$\boldsymbol{T}^* = \text{diag}(\boldsymbol{a}^t)\exp(-\mathbf{C}/\lambda)\text{diag}(\boldsymbol{b}^t) \tag{16}$$

where $t$ is the iteration and $\boldsymbol{a}^t = \boldsymbol{\mu}/\exp(-\mathbf{C}/\lambda)\boldsymbol{b}^{t-1}$ and $\boldsymbol{b}^t = \boldsymbol{\nu}/\exp(-\mathbf{C}/\lambda)\boldsymbol{a}^t$, with the initialization on $\boldsymbol{b}^0 = \boldsymbol{1}$. In our experiments, we used $t = 100$ and $\lambda = 0.1$ based on validation performance.

### D.2 Relaxed Marginal Constraints with Unbalanced Optimal Transport

Due to strict marginal constraints in Eq equation 14, optimal transport may be unrealistic in real-world scenarios where data distributions are noisy, incomplete, or unbalanced. The Unbalanced Optimal Transport (UoT) (Chizat et al., 2018; Liero et al., 2018) addresses this challenge by relaxing the marginal constraints, allowing for partial matching through penalties on mass creation or destruction. In particular, UoT solves

$$\boldsymbol{T}^* = \underset{\boldsymbol{T} \in \mathbb{R}^{MXN}}{\arg\min} \sum_{i=1}^{M}\sum_{j=1}^{N} \boldsymbol{T}_{ij}\boldsymbol{C}_{ij} - \lambda H(\boldsymbol{T}) \tag{17}$$

$$+ \rho_1 \text{KL}(\boldsymbol{T}\boldsymbol{1}^N||\boldsymbol{\mu}) + \rho_1 \text{KL}(\boldsymbol{T}^\top \boldsymbol{1}^M||\boldsymbol{\nu})$$

here, $\rho_1$ and $\rho_2$ represent the marginal regularization parameters, and $\text{KL}(\boldsymbol{P}||\boldsymbol{Q})$ denotes the Kullback-Leibler divergence between two positive vectors. Similar to the classical OT formulation, there are solvers based on the Sinkhorn algorithm that can address Eq. equation 18 (Pham et al., 2020). However, these solvers typically require more iteration steps to converge to optimal solutions due to the added complexity introduced by the relaxed marginal constraints.

## E Unbalance Optimal Transport (UoT)

Table 10: MGPATH(CLIP) performance and running time (in second) comparison between OT and UoT.

| Methods | TCGA-NSCLC | | | |
|---|---|---|---|---|
| | AUC ↑ | F1 ↑ | ACC ↑ | Time (s) ↓ |
| MGPATH(CLIP) (OT, 4 text prompts) | 76.2±2.2 | 69.0±3.5 | 69.3±2.8 | 1482 |
| MGPATH(CLIP) (UoT, 4 text prompts) | 77.0±1.8 | 70.2±3.4 | 70.4±3.3 | 3260 |
| | TCGA-RCC | | | |
| MGPATH(CLIP) (OT, 4 text prompts) | 92.1±2.8 | 76.5±5.2 | 81.7±2.9 | 1451 |
| MGPATH(CLIP) (UoT, 4 text prompts) | 92.8±2.4 | 76.8±4.7 | 82.4 ± 2.4 | 3049 |

To conduct a comparative evaluation of the performance of `MGPATH` using optimal transport versus unbalanced optimal transport (Section D.2) given the more flexible constraints in UoT, we conducted an additional experiment. To be specific, we test on the `TCGA-NSCLC` and `TCGA-RCC` datasets with `MGPATH(CLIP)` using a 4-text-prompt setting. Table 10 presents our findings where the running time is computed as seconds of average across five-folds. The results show that UoT outperforms OT with an approximate 1% improvement across all metrics. However, UoT is approximately 2 times slower than OT. This increase is attributed to the added flexibility and complexity introduced by relaxing the marginal constraints in the UoT formulation. Given this trade-off, we choose OT as the main distance in `MGPATH` and leave the UoT version for further evaluation. It is also important to know that our OT formulation leverages approximate solutions through the regularized formulation (Eq.,equation 15) and produces smoothed optimal mappings $\boldsymbol{T}^*$, which can implicitly help the model adapt to perturbations like UoT.

## F   Ablation Studies on Data Augmentation

To evaluate the effect of data augmentation on model performance, we conducted experiments comparing `MGPATH(PLIP-G)` with and without augmentation. In addition, we trained a variant, `MGPATH(PLIP-G)(COS)`, where the optimal transport module in the `MGPATH(PLIP-G)` architecture was replaced by cosine similarity. All experiments were carried out on the `TCGA-NSCLC` dataset under the 16-shot setting.

As illustrated in Table 11, the cosine similarity configuration exhibits unstable behavior, with a sharp decline in F1 and ACC when data augmentation is applied, indicating difficulty in adapting to heterogeneous data in the training data. In contrast, the optimal transport configuration not only achieves the best overall results but also remains consistently strong across augmented and non-augmented settings.

Table 11: Impact of data augmentation on the performance of `MGPATH` using cosine similarity and optimal transport configurations on the `TCGA-NSCLC` dataset.

| Method | TCGA-NSCL | | |
| --- | --- | --- | --- |
| | AUC | F1 | ACC |
| **MGPATH(PLIP-G)(COS)** | $91.13 \pm 3.20$ | $82.88 \pm 3.55$ | $83.05 \pm 3.53$ |
| - w/o augmentation | $92.92 \pm 3.88$ | $74.35 \pm 23.40$ | $78.17 \pm 15.93$ |
| **MGPATH(PLIP-G)** | $93.95 \pm 2.71$ | $88.53 \pm 4.29$ | $88.63 \pm 4.26$ |
| - w/o augmentation | $91.94 \pm 5.33$ | $84.49 \pm 6.02$ | $84.67 \pm 5.98$ |

## G   Training and Inference Efficiency

To assess the computational efficiency of `MGPATH(PLIP-G)` based on OT and cosine similarity, `MGPATH(PLIP-G)(OT)` and `MGPATH(PLIP-G)(COS)`, respectively. We conducted an additional experiments. To concreat, we evaluate training and inference using the metrics **Throughput (samples/sec)** on the dataset `TCGA-NSCLC` with text prompts generated by `Mistral Medium 3` (Mistral AI, 2024) in 16-shots setting.

Table 12 shows that although the optimal transport configuration requires more computation and yields lower training and inference throughput compared to cosine similarity, it consistently achieves higher accuracy. This highlights the strength of optimal transport in improving model generalization, as it is able to leverage complex data distributions more effectively despite the additional computational overhead.

Table 12: Comparison of average Training and Inference Throughput (samples/sec) on a single NVIDIA A100 GPU with only batch size 1 for `MGPATH(PLIP-G)(OT)` and `MGPATH(PLIP-G)(COS)`.

| Configuration | Training | Inference | |
| --- | --- | --- | --- |
| | Throughput (samples/sec)↑ | Throughput (samples/sec)↑ | ACC↑ |
| **MGPATH(PLIP-G)(COS)** | 16.00 0.29 | 31.59 0.32 | 83.05 3.53 |
| **MGPATH(PLIP-G)** | 10.67 2.52 | 19.05 5.59 | 88.63 4.26 |

