# OpenReview forum: "MGPATH: A Vision-Language Model with Multi-Granular Prompt Learning for Few-Shot Whole Slide Pathology Classification"
_TMLR — Accepted by TMLR_

### Review · Reviewer_gSYA · 2025-06-04

**Summary Of Contributions:**

This manuscript presents MGPATH, a vision-language model for whole‐slide image (WSI) classification. The authors adapt the Prov‐GigaPath foundation vision model into a vision‐language framework by aligning it with a pathology‐specific text encoder PLIP. They introduce a multi‐granular prompt learning strategy that (1) generates multiple learnable prompts at both low and high magnification levels, (2) fuses prompt embeddings with frozen visual features at the individual patch and patch‐group level, and (3) measures the distance between visual and text embeddings using an optimal transport metric. Experimental results on three TCGA datasets demonstrate consistent improvements over state‐of‐the‐art multiple‐instance‐learning and other vision‐language baselines.

**Audience:**

Yes

**Broader Impact Concerns:**

The authors do not provide a Broader Impact Statement. It might be helpful if the authors can briefly discuss the potential diagnostic misuse  and the generalizability issues.

**Claims And Evidence:**

Yes

**Requested Changes:**

In the ablation study, it would be beneficial if the authors can discuss the cases when alpha = 0 and alpha = 1. I am also not very sure what "without multi-granular" mean. Does it mean that only high-resolution patches and texts are used in training? I am curious about how the model would perform, if only high-resolution patches are used as the image input but align with both high and low-resolution textual embeddings.

It would be beneficial if the authors can include details such as patch size, magnification levels and computational costs.

Please replace the placeholder “??” on page 11 with the correct table number.

**Strengths And Weaknesses:**

Strength:

The manuscript Leverages the foundation model Prov‐GigaPath  trained on 1.3 billion pathology patches and fine-tune it with an additional 923K image–text pairs.

The authors propose to combine the patch‐level prompting and spatial patch group prompting,

The authors propose to utilize optimal transport (OT)‐based distance to address the limitations of cosine similarity.

MGPATH was tested on three TCGA datasets, demonstrating consistent improvements over state‐of‐the‐art multiple‐instance‐learning and other vision‐language baselines.

Weaknesses:

The high- and low-resolution descriptive prompts are generated using the GPT-4 model. Since GPT-4 is a general-purpose model, I am not sure whether the prompts are reliable. It is also unclear how much of the performance of MGPATH depends on the quality of the language model.

In the ablation study, the authors do not discuss the case when alpha = 0. I am suspicious about whether the spatial patch group-based prompting is helpful.

The authors do not provide sufficient details in the experiments. I cannot find the patch size and the magnification levels that define the high- and low-resolution images. It would be helpful if the authors can discuss training and inference time of the proposed model.

---

> ### Author Response · Authors · 2025-07-17
> **Answer Questions 1 and 2 of Reviewer gSYA**
>
> **Q1. The reviewer expresses concern about the reliability of using GPT-4—a general-purpose model—to generate high- and low-resolution descriptive prompts. The reviewer also question how much of MGPATH’s performance depends on the quality of the language model.**
>
> **Response**: We want to thank the reviewer for the valuable comment. All the generated text prompts underwent human verification to assert they include visual description relevant to the cancer subtype in the class description. To further verify the impact of different LLMs on MGPATH performance, we conducted experiments with text prompts generated by various LLMs, including Deepseek R1, Grok-3, GPT-o3, Gemini pro 2.5 and Mistral Medium. The obtained results are presented in the response of Reviewer xKaA, validating the robustness of MGPATH. We will add this experiment to the subsection Effect of Large Language Models under Section Ablation Studies.
>
> **Q2. The reviewer suggests that the ablation study should include a discussion of the cases where α = 0 and α = 1. They also seek clarification on the term “without multi-granular,”**
>
> **Response**: We thank you Reviewer for your question. We want to clarify that our multi-granular attention = (i) regional attention +  (ii) individual patch-level attention. So the alpha = 0 or 1 indicates the contribution of the first term (i). Thus, the term **without multi-granular** corresponses to the case alpha = 0. This is not about resolution.

---

> ### Author Response · Authors · 2025-07-17
> **Answer Questions 3 and 4 of Reviewer gSYA**
>
> **Q3. The reviewer is curious about the model’s performance when only high-resolution image patches are used as input, but are aligned with both high- and low-resolution textual embeddings.**
>
> **Response**: we would like to thank the reviewer for the insightful suggestion. we conducted an experiment where MGPATH(PLIP-G) utilized only high-resolution image patches while still aligning with both high- and low-resolution textual embeddings. This experiment was performed under the 16-shot setting on the TCGA-NSCLC dataset. The results are presented in the table below:
>
> | Configurations             | AUC         | F1           | ACC          |
> |----------------------------|-------------|--------------|--------------|
> | only high resolution       | 89.48 ± 3.85 | 67.78 ± 16.37   | 72.08 ± 12.12   |
> | **high + low resolution**       | **93.02 ± 2.99** | **84.64 ± 4.75**   | **84.77 ± 4.67**   |
>
> We observed a significant drop in performance when low-resolution patches were excluded. This highlights the importance of global contextual information, such as the overall tissue and cellular architecture which are captured at low magnification for improving model performance.
>
> **Q4. The reviewer recommends including additional implementation details—such as patch size, magnification levels, and computational costs to improve the clarity and reproducibility of the work.**
>
> **Response**: We would appreciate the valuable suggestion. We will add a paragraph under the subsection “Implementation Detail” to describe the patch size, magnification level and computing requirement for developing MGPATH. To clarify, MGPATH leverages dual magnifications 5x and 10x. The patches are cropped to a size of 256x256. In the appendix, we also described other relevant hyper-parameters and number of epochs.

---

> ### Author Response · Authors · 2025-07-17
> **Answer Questions 5 and 6 of Reviewer gSYA**
>
> **Q5. The reviewer points out that there is a placeholder “??” on page 11 and requests that it be replaced with the correct table number.**
>
> **Response**: We would like to apologize for any confusion caused by this missing reference. We will update the manuscript to fix this problem.
>
> **Q6. The reviewer notes the absence of a Broader Impact Statement and suggests that the authors briefly address potential diagnostic misuse and generalizability concerns.**
>
> **Response**: We thank the reviewer for the valuable comment. We added the Broader Impact Statement section in our manuscript. To clarify, MGPATH is developed to investigate the feasibility of vision-language models (VLM) to pathology under few-shot learning, simulation scenarios of rare diseases with very limited available training samples. However, the model is not intended for clinical usage as clinical support or diagnostic tool. The model is not designed to be substituted for professional medical diagnosis.

---

### Review · Reviewer_MdBd · 2025-06-08

**Summary Of Contributions:**

The paper introduces MGPath, a novel vision-language model that tries to address problems in few-shot whole-slide pathology classification. Key contributions include (i) extending Prov-GigaPath into a vision-language model using PLIP text encoder; (ii) proposing a multi-granular attention mechanism; and (iii) employing optimal transport as a robust alignment metric between visual and text embeddings instead of cosine similarity. Based on the experiments provided, the method shows performance improvements across 3 public datasets compared to MIL methods and other vision-language approaches.

**Audience:**

Yes

**Broader Impact Concerns:**

No concern.

**Claims And Evidence:**

No

**Requested Changes:**

I'd certainly recommend revising the text for improved clarity, structure, and consistency. Above, I have outlined specific weaknesses observed in the current version of the paper and my suggestions for change or improvement.

**Strengths And Weaknesses:**

**Strength:**
The paper presents a novel and well-motivated approach for few-shot whole slide image classification, introducing multi-granular prompt learning and optimal transport alignment. It is further backed by many experiments across multiple datasets and model backbones.

**Weaknesses:**

1. From an editorial standpoint, the paper requires substantial revision. For example, the Introduction section includes excessive technical detail that is more appropriate for the Methods section (specifically, the last three paragraphs). Additionally, inconsistent and duplicative terminology is used throughout the paper and tables—for instance, “GigaPath-PLIP” and “PLIP-G” appear to refer to the same model variant and should be unified for clarity. Important details are also missing from the main text: the number of shots used in few-shot experiments is mentioned only in the appendix, and a table reference is missing in Section 4.4.

2. The few-shot experiments lack evaluations across a range of shot numbers, which is important to assess the model’s robustness under varying levels of supervision. Given the paper’s emphasis on both zero-shot and few-shot learning, this omission is important, especially since, in real-world clinical settings, few-shot scenarios often involve rare cancers with only a handful of annotated cases. I recommend evaluating the model using a broader range of shots (e.g., 1 to 32). For reference, consider the methodology in this paper: https://arxiv.org/pdf/2411.19666

3. Figure 1 lacks clarity and is not sufficiently intuitive. Even after reading the caption, the meaning of “t+1”, “t+2”, etc., and their relation to patches and prompts remains unclear to me, particularly when comparing the two rows in the figure and the presence of different red square boxes in each patch grid. A clearer and more informative visual illustration would greatly improve reader comprehension.

4. It appears that the authors pretrained different variations of their model using the 923K image-text pairs, while the baseline models were not pretrained on this dataset. If this is correct, the comparison may not be entirely fair due to differing pretraining data sources. In this case, I recommend including experiments where MGPath, ViLa-MIL, and MSCPT are pretrained on a comparable dataset to ensure a fair and meaningful comparison. This will emphasize whether the multi-granular attention is the main source of performance improvement or the extra training data.

5. The authors also claim that MGPath is parameter-efficient; however, the number of parameters reported for the full model (MGPath with PLIP-G at 5.35M; Table 1) is actually higher than many baseline models it is compared against. This raises questions about the basis of the parameter efficiency claim, and the authors should clarify what exactly they mean.

---

> ### Author Response · Authors · 2025-07-17
> **Answer The Questions 1 and 2 of Reviewer MdBd**
>
> **Q1. The reviewer notes that the paper uses inconsistent and duplicative terminology-such as “GigaPath-PLIP” and “PLIP-G”-to refer to the same model variant, and recommends unifying the naming for clarity.**
>
> **Response**: We apologize for the confusion caused by inconsistent terminology. The terms “GigaPath-PLIP” and “PLIP-G” both refer to the same variant of our MGPATH model, which aligns PLIP’s text encoder with the vision encoder from Prov-GigaPath. To maintain consistency and improve clarity, we have standardized the naming throughout the paper and now refer to this variant exclusively as “PLIP-G.” We have also thoroughly reviewed and unified the abbreviations used for all MGPATH variants to avoid further ambiguity.
>
> **Q2. The reviewer points out that important details are missing from the main text-for example, the number of shots used in few-shot experiments is only mentioned in the appendix, and Section 4.4 lacks a reference to the relevant table.**
>
> **Response**: We thank the reviewer for the valuable comment and apologize for the oversight. The number of shots used in the few-shot experiments was indeed only mentioned in the appendix. We will revise the main text to include this important detail alongside the interpretation of the experimental results. Additionally, we will correct the missing table reference in Section 4.4 to improve clarity and completeness.

---

> ### Author Response · Authors · 2025-07-17
> **Answer The Questions 3 and 4 of Reviewer MdBd**
>
> **Q3. The reviewer highlights that the study does not evaluate performance across a range of shot numbers, which is important for assessing model robustness in practical scenarios**
>
> **Response**: We thank the reviewer for this valuable suggestion. To assess MGPATH’s robustness under different supervision levels, we have run additional few-shot experiments with K = 1, 2, 4, 8, and 16 annotated cases per class, following the protocol recommended in Tong Ding et al. (2024). The new results will now be presented in the subsection “Evaluating MGPATH (PLIP-G) with Varying Shot Numbers” - show that MGPATH (PLIP-G) consistently outperforms all baselines across the entire range of K. These findings further confirm the model’s robustness in real-world clinical scenarios where annotated samples may be extremely scarce. We have also cited the reviewer’s referenced paper to acknowledge the methodology that motivated this evaluation. We would like to enclose a figure to describe the experimental result in the following link.
>
> https://anonymous.4open.science/r/MGPATH-TMLR--6890/tcga_brca_k_shots.png
>
> **Q4. The reviewer finds Figure 1 unclear and unintuitive. They are confused by the meaning of “t+1,” “t+2,” the dot patches, prompts, and the differing red square boxes between rows. The reviewer suggest providing a clearer and more informative visual illustration to enhance reader understanding.**
>
> **Response**: We thank the reviewer for the helpful feedback regarding Figure 1. We acknowledge that the original figure and caption did not clearly convey the meaning of the tokens (e.g., “t+1”, “t+2”) or the purpose of the red boxes. To address this, we have revised both the figure and its caption for greater clarity. In the updated caption, we clarify that tokens such as t+1, t+2, and t+3 correspond to spatially adjacent patches within a larger tissue region (highlighted by the larger red box), which together contribute to the model’s diagnostic decision. In contrast, t+9 represents a more distant or isolated patch from a different region (highlighted by the smaller red box), which also plays a role in the prediction. These annotations help illustrate how our method integrates both local and distant contextual information through multi-granular attention.
>
>
> Furthermore, we also highlight that our multi-granular attention mechanism improves over standard self-attention, which typically attends to tokens such as t+3 and t+9 independently, based on their individual relevance scores to the prompts. In contrast, we explicitly incorporate both regional attention (rewarding nearby, semantically connected tokens, e.g., t+1, t+2, t+3) and individual patch-level attention  (e.g., t+9). This results in richer prompt-guided slide features that better capture the underlying pathology. The updated figure and caption will be included in the revised manuscript to enhance reader comprehension.

---

> ### Author Response · Authors · 2025-07-17
> **Answer The Questions 5 and 6 of Reviewer MdBd**
>
> **Q5: The reviewer notes that MGPATH variants were pretrained on 923K image-text pairs, while baseline models were not, potentially leading to an unfair comparison.**
>
> **Response**: We thank the reviewer for this important observation. To clarify, MGPATH was evaluated across multiple architectures, including CLIP-based, PLIP-based, and our proposed PLIP-G variant. Among these, only PLIP-G was further pre-trained on the additional dataset of 923K image-text pairs. The other variants (CLIP and PLIP) were not exposed to this extra data, ensuring a fair comparison with baselines such as ViLa-MIL and MSCPT.
>
> Our experiments demonstrate that even when operating under the same architectures (CLIP and PLIP), MGPATH consistently outperforms these baselines. This supports our claim that the performance gains primarily stem from the proposed multi-granular attention mechanism, rather than from extra samples in pretraining data.
>
> We will revise the manuscript to clarify this point more explicitly and ensure the experimental setup is clearly described for readers.
>
> **Q6: The reviewer questions the authors’ claim that MGPATH is parameter-efficient, noting that the reported parameter count (5.35M for MGPATH with PLIP-G) is higher than that of several baseline models. The reviewer request clarification on what the authors mean by parameter efficiency.**
>
> **Response**: We thank the reviewer for the insightful question. The total number of parameters in MGPATH(PLIP-G) comprises three main components: the learnable parameters (5,250,048), the PLIP text encoder (37,929,985), and Prov-GigaPath (1,134,953,984). However, it is important to note that both the PLIP text encoder and Prov-GigaPath are frozen in our method and are not updated during training. Among the learnable parameters, the majority (4,198,400) are used to project the outputs of the PLIP text encoder and Prov-GigaPath into a shared feature space, primarily due to the large output size of Prov-GigaPath. Overall, the learnable parameters constitute only 0.45% of the total model parameters.

---

### Review · Reviewer_xKaA · 2025-07-07

**Summary Of Contributions:**

This work developed MGPATH, a vision-language model for whole slide images. It aligns the encoder of a foundational vision model (Prov-GigaPath) with the text encoder of a foundational vision-language model (PLIP) using lightweight adaptors trained on 923K image-text pairs. This vision-language model is then used to conduct few-shot WSI classification with multi-granular prompt learning. Extensive experiments show that it outperforms state-of-the-art methods.

**Audience:**

Yes

**Broader Impact Concerns:**

There are no concerns on the ethical implications.

**Claims And Evidence:**

Yes

**Requested Changes:**

See the weakness.

**Strengths And Weaknesses:**

Strengths
- The lightweight adaptor training strategy is efficient and elegant.
- The idea of multi-granular prompt learning is insightful and novel.
- The experiments are comprehensive and carefully designed.

Weakness
- Although the presentation is generally clear, some of the abbreviations and notations need clarification. I understand that MGPath has different versions depending on the CLIP backbone choices. Yet, can the authors clearly explain what MGPath, MGPath (ViT), MGPath (PILP), and MGPath (PLIP-G) are, preferably list all the different versions together, and explain their differences?
- It's not very clear to me how the few-shot classification experiments are conducted on the comparative methods, especially for the vision-language models CONCH and QULTT. Are they fine-tuned with training data, or is there a classifier built on the vision encoder?
- The authors use GPT-4 to generate a visual descriptive text prompt. Are there any quality controls on the GPT's output (i.e., evaluated by experts/consistency of responses)?

---

> ### Author Response · Authors · 2025-07-17
> **Answer Questions of Reviewer xKaA**
>
> **Q1.The reviewer requests a clear explanation of the abbreviations of MGPATH (e.g., MGPATH, MGPATH(ViT), MGPATH(PLIP), and MGPATH(PLIP-G)).**
>
> **Response**: We sincerely thank the reviewer for the valuable feedback. In this study, we release several variants for our proposed method MGPATH based on different CLIP-based backbones. Specifically, MGPATH(CLIP), and MGPATH(ViT) refer to versions based on CLIP-ResNet50 and CLIP-ViT-B/16 backbones, respectively. Those CLIP backbones are pretrained on general vision images. MGPATH(PLIP) denotes the variant using PLIP backbone, which is pre-trained on pathology images. Finally, MGPATH(PLIP-G) represents our strongest version, which aligns PLIP’s text encoder with the vision encoder from Prov-GigaPath. We apologize for any confusion. We already updated our manuscript with a clear explanation for the abbreviations of MGPATH.
>
> **Q2. The reviewer is asking for clarification on how few-shot classification is conducted for the comparative methods, particularly for the vision-language models CONCH and QUILT.**
>
> **Response**: We would like to thank the reviewer. In our manuscript, the experiments in Tables 1 and 2 were conducted under 16-shot settings. For CONCH and QUILT, we followed the standard practice in prior works by using frozen vision encoders to extract image feature embeddings and fine-tuning only the text encoders. We will update the manuscript to include this clarification, along with supporting experimental results, to avoid any confusion.
>
> **Q3. The reviewer asks whether any quality control measures were applied to the GPT-4-generated visual descriptive text prompts-such as expert evaluation or checks for consistency in the responses.**
>
> **Response**: We thank the reviewer for this insightful comment. All generated text prompts were manually verified to ensure they included visual descriptions relevant to the cancer subtype in the class definitions. To further assess the impact of different LLMs on MGPATH's performance, we conducted additional experiments using prompts generated by various LLMs, including DeepSeek-VL R1, Grok-3, GPT-3.5, Gemini Pro 2.5, and Mistral Medium. We will include a subsection Effect of Large Language Models under Ablation Studies, demonstrate the robustness of MGPATH. As described in the following table, we observe that prompts from other state-of-the-art LLMs yield comparable or even improved performance, validating the consistency and adaptability of MGPATH across different LLM-generated descriptions.
>
>
> ### Table: Ablation experiments of input text prompts from different LLM (TCGA-BRCA)
>
> | Methods                         | AUC               | F1                | ACC               |
> |----------------------------------|--------------------|--------------------|--------------------|
> | GPT-4o                       | 93.02 ± 2.99         | 84.64 ± 4.75       | 84.77 ± 4.67         |
> | Grok-3                       | 93.07 ± 3.56         | 85.50 ± 4.53       | 85.58 ± 4.51         |
> | GPT-o3                       | `_93.87 ± 3.31_`   | 86.19 ± 5.28       | 86.29 ± 5.26         |
> | Gemini-2.5-pro               | 93.17 ± 3.12         | 86.72 ± 3.50       | 86.80 ± 3.48         |
> | Deepseek-R1                  | 93.40 ± 2.80         | `_86.87 ± 4.68_` | `_87.01 ± 4.63_`   |
> | **Mistral Medium 3**         | **93.95 ± 2.71**     | **88.53 ± 4.29**   | **88.63 ± 4.26**     |

---

### Decision · Action_Editor_h89b · 2025-08-26

**Recommendation:** Accept with minor revision

**Additional Comments:**

The paper proposes a model called MGPATH, a pathology-focused vision–language model that aligns a vision encoder with a text encoder via lightweight adapters. It offers a couple of nice innovations: a multi-resolution style approach (patch- and region-level prompting across magnifications) and the use of optimal transport for vision–text alignment. On several datasets, it outperforms strong baselines. The authors also provide ablations supporting the design choices.

Reviewers generally liked the adapter strategy, the multi-resolution idea, and the solid results. The main concerns were presentation clarity, missing details (for example, on few-shot protocol and patch size/magnifications), and some requests for additional experiments (like a sweep over shots). The authors have added these, or promised to.

The paper is on track to be accepted; in the revision I would additionally like to see, along with the other items the authors promised, a runtime comparison for OT versus cosine similarity (and perhaps a more general description of runtime/compute costs, which may end up being a downside that should be noted). A description of the manual verification protocol would be great.

As an additional minor example to beef up evidence for, the authors mention the issue with data augmentation (“mitigating perturbations that might occur during the data augmentation process”) to motivate OT; is there an explicit way to show that this is where the OT benefit comes from, e.g., by comparing cos vs OT on a dataset with no augmentation vs with augmentations?

**Audience:**

Yes

**Audience Explanation:**

The work is in the mainstream of multimodal models and therefore of interest to the TMLR audience.

**Claims And Evidence:**

Yes

**Claims Explanation:**

There is solid evidence for almost all the claims made in the paper.

---

> ### Author Response · Authors · 2025-09-25
> **Preparing for the Camera-ready version**
>
> Dear Action Editor,
>
> Thank you very much for your valuable feedback. We have made the following changes in the camera-ready version of our paper:
>
> 1) In the appendix section, adding ablation studies on the effectiveness of data augmentation for two configurations, cosine similarity and optimal transport, on the performance of MGPATH(PLIP-G).
>
> 2) In the appendix section, we add ablation studies to investigate the running time of MGPATh(PLIP-G) with two configurations: cosine similarity and optimal transport.
>
> 3) In the Ablation Study section, include the figures of the results for additional few-shots experiments with K=2, 4, 6, 8, 16.
>
> 4) Update the manuscript to correct the abbreviations of MGPATH. Adding an explanation to explain the abbreviations MGPATH (PLIP-G), MGPATH(PLIP), and MGPATH(ViT).
>
> 5) Provide ablation experiments to investigate the performance of MGPATH(PLIP-G) with text-prompts generated by various large language models, such as GPT-4o, or Grok-3.
>
> 5) Publicly released the code repository.
>
> Once again, we sincerely appreciate the comments from the reviewers and the Action Editor, which have greatly enhanced the quality of our work.
>
> Best wishes,
>
> MGPATH team.